# Improving Probabilistic Diffusion Models With Optimal Diagonal Covariance Matching

**Zijing Ou**[1,*], **Mingtian Zhang**[2,*], **Andi Zhang**[3], **Tim Z. Xiao**[456], **Yingzhen Li**[1], **David Barber**[2]
[1]Imperial College London, [2]University College London, [3]University of Manchester,
[4]Max Planck Institute for Intelligent Systems, Tubingen [5]University of Tubingen, [6]IMPRS-IS.
z.ou22@imperial.ac.uk   m.zhang@cs.ucl.ac.uk

## Abstract

The probabilistic diffusion model has become highly effective across various domains. Typically, sampling from a diffusion model involves using a denoising distribution characterized by a Gaussian with a learned mean and either fixed or learned covariances. In this paper, we leverage the recently proposed covariance moment matching technique and introduce a novel method for learning the diagonal covariance. Unlike traditional data-driven diagonal covariance approximation approaches, our method involves directly regressing the optimal diagonal analytic covariance using a new, unbiased objective named Optimal Covariance Matching (OCM). This approach can significantly reduce the approximation error in covariance prediction. We demonstrate how our method can substantially enhance the sampling efficiency, recall rate and likelihood of commonly used diffusion models.

## 1 Introduction

Diffusion models (Sohl-Dickstein et al., 2015; Ho et al., 2020; Song & Ermon, 2019) have achieved remarkable success in modeling complex data across various domains (Rombach et al., 2022; Li et al., 2022; Poole et al., 2022; Ho et al., 2022; Hoogeboom et al., 2022; Liu et al., 2023). A conventional diffusion model operates in two stages: a forward noising process, indexed by $t \in [1, T]$, which progressively corrupts the data distribution into a Gaussian distribution via Gaussian convolution, and a reverse denoising process, which generates images by gradually transforming Gaussian noise back into coherent data samples. In traditional diffusion models, the generation process typically predicts only the mean of the denoising distribution while using a fixed, pre-defined variance (Ho et al., 2020). This approach often requires a very large number of steps, $T$, to produce high-quality, diverse samples or to achieve reasonable model likelihoods, leading to inefficiencies during inference.

To address this inefficiency, several works have proposed estimating the diagonal covariance of denoising distributions rather than relying on pre-defined variance values. For instance, Bao et al. (2022b) introduces an analytical form of isotropic, state-independent covariance that can be estimated from the data. This analytical solution achieves an optimal form under the constraints of isotropy and state independence. A more flexible approach involves learning a state-dependent diagonal covariance. Nichol & Dhariwal (2021) propose learning this form by optimizing the variational lower bound (VLB). Bao et al. (2022a) explore learning a state-dependent diagonal covariance directly from data, also examining its analytical form. These methods have demonstrated improved image quality with fewer denoising steps. Learning the covariance through VLB optimization (Nichol & Dhariwal, 2021) has become a widely adopted strategy in state-of-the-art image and video generative models.

Building on this line of research, the goal of our paper is to develop an improved denoising covariance strategy that enhances both the generation quality and likelihood evaluation while reducing the number of total time steps. Recently, Zhang et al. (2024b) derived the optimal state-dependent full covariance for the denoising distribution. While this method offers greater flexibility than state-dependent diagonal covariance, it requires $O(D^2)$ storage for the Hessian matrix and $O(D)$ network evaluations per denoising step, where $D$ is the data dimension. This makes it impractical for high-dimensional applications, such as image generation. To address this limitation, we propose a novel,

---

*Equal contribution. Code is available at: https://github.com/J-zin/OCM_DPM.

unbiased covariance matching objective that enables a neural network to match the diagonal of the optimal state-dependent diagonal covariance. Unlike previous methods (Bao et al., 2022a; Nichol & Dhariwal, 2021), which learn the diagonal covariance directly from the data, our approach estimates the diagonal covariance from the learned score function. We show that this method significantly reduces covariance estimation errors compared to existing techniques. Moreover, we demonstrate that our approach can be applied to both Markovian (DDPM) and non-Markovian (DDIM) diffusion models, as well as latent diffusion models. This results in improvements in generation quality, recall, and likelihood evaluation, while also reducing the number of function evaluations (NFEs).

## 2 BACKGROUND OF PROBABILISTIC DIFFUSION MODELS

We first introduce two classes of diffusion models that will be explored further in our paper.

### 2.1 MARKOVIAN DIFFUSION MODEL: DDPM

Let $q(x_0)$ be the true data distribution, denoising diffusion probabilistic models (DDPM) (Ho et al., 2020) constructs the following Markovian process

$$q(x_{0:T}) = q(x_0) \prod_{t=1}^{T} q(x_t|x_{t-1}), \tag{1}$$

where $q(x_t|x_{t-1}) = \mathcal{N}(\sqrt{1-\beta_t}x_{t-1}, \beta_t I)$ and $\beta_{1:T}$ is the pre-defined noise schedule. We can also derive the following skip-step noising distribution in a closed form

$$q(x_t|x_0) = \mathcal{N}(\sqrt{\bar{\alpha}_t}x_0, (1-\bar{\alpha}_t)I), \tag{2}$$

where $\bar{\alpha}_t \equiv \prod_{s=1}^{t}(1-\beta_s)$. When $T$ is sufficiently large, we have the marginal distribution $q(x_T) \to \mathcal{N}(0, I)$. The generation process utilizes an initial distribution $q(x_T)$ and the denoising distribution $q(x_{t-1}|x_t)$. We assume for a large $T$ and variance preversing schedule discussed in Equation 1, we can approximate $q(x_T) \approx p(x_T) = \mathcal{N}(0, I)$. The true $q(x_{t-1}|x_t)$ is intractable and is approximated as a variational distribution $p_\theta(x_{t-1}|x_t)$ within a Gaussian family, which defines the reverse denoising process as

$$p_\theta(x_{t-1}|x_t) = \mathcal{N}(x_{t-1}|\mu_{t-1}(x_t; \theta), \Sigma_{t-1}(x_t; \theta)). \tag{3}$$

With Tweedie's Lemma (Efron, 2011; Robbins, 1992), we can have the following score representation

$$\mu_{t-1}(x_t; \theta) = (x_t + \beta_t \nabla_{x_t} \log p_\theta(x_t))/\sqrt{1-\beta_t}, \tag{4}$$

where the approximated score function $\nabla_{x_t} \log p_\theta(x_t) \approx \nabla_{x_t} \log q(x_t)$ can be learned by the denoising score matching (DSM) (Vincent, 2011; Song & Ermon, 2019).

For the covariance $\Sigma_{t-1}(x_t; \theta)$, two heuristic choices were proposed in the original DDPM paper: 1. $\Sigma_{t-1}(x_t; \theta) = \beta_t$, which is equal to the variance of $q(x_t|x_{t-1})$ and 2. $\Sigma_{t-1}(x_t; \theta) = \tilde{\beta}_t$ where $\tilde{\beta}_t = (1-\bar{\alpha}_{t-1})/(1-\bar{\alpha}_t)\beta_t$ is the variance of $q(x_{t-1}|x_t, x_0)$. Although heuristic, Ho et al. (2020) show that when $T$ is large, both options yield similar generation quality.

### 2.2 NON-MARKOVIAN DIFFUSION MODEL: DDIM

In additional to the Markovian diffusion process, denoising diffusion implicit models (DDIM) (Song et al., 2020) only defines the condition $q(x_t|x_0) = \mathcal{N}(\sqrt{\bar{\alpha}_t}x_0, (1-\bar{\alpha}_t)I)$ and let

$$q(x_{t-1}|x_t) \approx \int q(x_{t-1}|x_t, x_0)p_\theta(x_0|x_t)dx_0, \tag{5}$$

where $q(x_{t-1}|x_t, x_0) = \mathcal{N}(\mu_{t-1}, \sigma_{t-1}^2)$ with

$$\mu_{t-1} = \sqrt{\bar{\alpha}_{t-1}}x_0 + \sqrt{1-\bar{\alpha}_{t-1}-\sigma_{t-1}^2}(x_t - \sqrt{\bar{\alpha}_t}x_0)/\sqrt{1-\bar{\alpha}_t}. \tag{6}$$

When $\sigma_{t-1} = \sqrt{(1-\bar{\alpha}_{t-1})/(1-\bar{\alpha}_t)\beta_t}$, the diffusion process becomes Markovian and equivalent to DDPM. Specifically, DDIM chooses $\sigma \to 0$, which implicitly defines a non-Markovian diffusion

process. In the original paper (Song et al., 2020), the $q(x_0|x_t)$ is heuristically chosen as a delta function $p_\theta(x_0|x_t) = \delta(x_0 - \mu_0(x_t; \theta))$ where

$$\mu_0(x_t; \theta) = (x_t + (1 - \bar{\alpha}_t)\nabla_{x_t} \log p_\theta(x_t))/\sqrt{\bar{\alpha}_t}. \tag{7}$$

In both DDPM and DDIM, the covariance of $p_\theta(x_{t-1}|x_t)$ or $p_\theta(x_0|x_t)$ are chosen based on heuristics. Nichol & Dhariwal (2021) have shown that the choice of covariance makes a big impact when $T$ is small. Therefore, for the purpose of accelerating the diffusion sampling, our paper will focus on how to improve the covariance estimation quality in these cases. We will first introduce our method in the next section and then compare it with other methods in Section 4.

## 3 DIFFUSION MODELS WITH OPTIMAL COVARIANCE MATCHING

Recently, Zhang et al. (2024b) introduce the optimal covariance form of the denoising distribution $q(x|\tilde{x}) \propto q(\tilde{x}|x)q(x)$ for the Gaussian convolution $q(\tilde{x}|x) = \mathcal{N}(x, \sigma^2 I)$, which can be seen as a high-dimensional extension of the second-order Tweedie's Lemma (Efron, 2011; Robbins, 1992). We further extend the formula to **scaled** Gaussian convolutions in the following theorem.

**Theorem 1** (Generalized Analytical Covariance Identity). *Given a joint distribution $q(\tilde{x}, x) = q(\tilde{x}|x)q(x)$ with $q(\tilde{x}|x) = \mathcal{N}(\alpha x, \sigma^2 I)$, then the covariance of the true posterior $q(x|\tilde{x}) \propto q(x)q(\tilde{x}|x)$, which is defined as $\Sigma(\tilde{x}) = \mathbb{E}_{q(x|\tilde{x})}[x^2] - \mathbb{E}_{q(x|\tilde{x})}[x]^2$, has a closed form:*

$$\Sigma(\tilde{x}) = \left(\sigma^4 \nabla_{\tilde{x}}^2 \log q(\tilde{x}) + \sigma^2 I\right)/\alpha^2. \tag{8}$$

See Appendix A.1 for a proof. This covariance form can also be shown as the optimal covariance form under the KL divergences (Bao et al., 2022b; Zhang et al., 2024b). We can see that the exact covariance in this case only depends on the score function, which indicates the exact covariance can be derived from the score function. In general, the score function already contains all the information of the denoising distribution $q(x|\tilde{x})$, see Zhang et al. (2024b) for further discussion. We here only consider the covariance for simplicity.

In the case of the diffusion model, we use the learned score function as a plug-in approximation in Equation (8). Although the optimal covariance can be directly calculated from the learned score function, but it requires calculating the Hessian matrix, which is the Jacobian of the score function. This requires $O(D^2)$ storage and $D$ number of network evaluation (Martens et al., 2012) for each denoising step at the time $t$. Zhang et al. (2024b) propose to use the following consistent diagonal approximation (Bekas et al., 2007) to remove the $O(D^2)$ storage requirement

Table 1: Comparison of different covariance choices on CIFAR10 (CS). We use Rademacher estimator ($M = 100$) for the diagonal covariance approximation in OCM-DDPM.

| | FiD ↓ | | | | NLL ↓ | | | |
|---|---|---|---|---|---|---|---|---|
| # timesteps | 5 | 10 | 15 | 20 | 5 | 10 | 15 | 20 |
| DDPM, $\tilde{\beta}$ | 58.28 | 34.76 | 24.02 | 19.00 | 203.29 | 74.95 | 44.94 | 32.20 |
| DDPM, $\beta$ | 254.07 | 205.31 | 149.67 | 109.81 | 7.33 | 6.51 | 6.06 | 5.77 |
| OCM-DDPM | **38.88** | **21.60** | **13.35** | **9.75** | **6.82** | **4.98** | **4.62** | **4.43** |

$$\text{diag}(H(\tilde{x})) \approx 1/M \sum_{m=1}^{M} v_m \odot H(\tilde{x})v_m, \tag{9}$$

where $H(\tilde{x}) \equiv \nabla_{\tilde{x}}^2 \log q(\tilde{x})$ and $v_m \sim p(v)$ is a Rademacher random variable (Hutchinson, 1990) with entries $\pm 1$ and $\odot$ denotes the element-wise product. In Table 1, we compare the generation quality of DDPM using different covariance choices. The results demonstrate that generation quality improves significantly when using the Rademacher estimator to estimate the optimal covariance, compared to the heuristic choices of $\beta$ and $\tilde{\beta}$. However, as shown in Figure 5 in the Appendix, achieving a desirable approximation on the CIFAR10 dataset necessitates $M \geq 100$ Rademacher samples. Each calculation of $v_m \odot H(\tilde{x})v_m$ requires a forward pass and a backward propagation, leading to roughly $2M$ network evaluations in total. This significantly slows down the generation speed, making it impractical for diffusion models. Inspired by Nichol & Dhariwal (2021); Bao et al. (2022a) and also the amortization technique used in variational inference (Kingma & Welling, 2013; Dayan et al., 1995), *we propose to use a network to match the diagonal Hessian, which only requires one network pass to predict the diagonal Hessian in the generation time and can be done in parallel with the score/mean predictions with no extra time cost.* In the next section, we introduce a novel unbiased objective to learn the diagonal Hessian from the learned score, which improves the covariance estimation accuracy and leads to better generation quality and higher likelihood estimations.

---

**Algorithm 1** Sampling procedure from $t \to t'$ in OCM-DDPM

1: Compute $\mu_{t'}(x_t)$ with Equation (15);
2: Compute $\Sigma_{t'}(x_t)$ with Equation (16);
3: Sample $\epsilon \sim \mathcal{N}(0, I)$;
4: Let $x_{t'} \leftarrow \mu_{t'}(x_t) + \Sigma_{t'}(x_t)^{1/2}\epsilon$.

---

**Algorithm 2** Sampling procedure from $t \to t'$ in OCM-DDIM

1: Compute $\mu_0(x_t)$ with Equation (18);
2: Compute $\Sigma_0(x_t)$ with Equation (19);
3: Sample $x_0 \sim \mathcal{N}(\mu_0(x_t), \Sigma_0(x_t))$;
4: Let $x_{t'} \leftarrow \sqrt{\bar{\alpha}_{t'}}x_0 + \sqrt{1 - \bar{\alpha}_{t'}}\frac{x_t - \sqrt{\bar{\alpha}_t}x_0}{\sqrt{1-\bar{\alpha}_t}}$.

---

## 3.1 UNBIASED OPTIMAL COVARIANCE MATCHING

To train a network $h_\phi(\tilde{x})$ to match the Hessian diagonal $\text{diag}(H(\tilde{x}))$, a straightforward solution is to directly regress Equation 9 for all the noisy data

$$\min_\phi \mathbb{E}_{q(\tilde{x})}||h_\phi(\tilde{x}) - \frac{1}{M}\sum_{m=1}^{M} v_m \odot H(\tilde{x})v_m||_2^2, \tag{10}$$

where $v_m \sim p(v)$. Although this objective is consistent when $M \to \infty$, it will introduce additional bias when $M$ is small. To avoid the bias, we propose the following unbiased optimal covariance matching (OCM) objective

$$L_{\text{ocm}}(\phi) = \mathbb{E}_{q(\tilde{x})p(v)}||h_\phi(\tilde{x}) - v \odot H(\tilde{x})v||_2^2, \tag{11}$$

which does not include an expectation within the non-linear L2 norm. The following theorem shows the validity of the proposed OCM objective.

**Theorem 2** (Validity of the OCM objective). *The objective in Equation* (11) *upper bounded the base objective (i.e., Equation* (10) *with $M \to \infty$). Moreover, it attains optimal when $h_\phi(\tilde{x}) = diag(H(\tilde{x}))$ for all $\tilde{x} \sim q(\tilde{x})$.*

See the Appendix A.2 for a proof. The integration over $v$ in Equation (11) can be unbiasedly approximated by the Monte-Carlo integration given $M$ Rademacher samples $v_m \sim p(v)$. In practice, we found $M = 1$ works well (see Table 11 in the appendix for the ablation study on varying $M$ values), which also shows the training efficiency of the proposed OCM objective. The learned $h_\phi$ can form the covariance approximation

$$\Sigma(\tilde{x}; \phi) = (\sigma^4 h_\phi(\tilde{x}) + \sigma^2 I)/\alpha^2. \tag{12}$$

We then discuss how to apply the learned covariance to diffusion models.

## 3.2 DIFFUSION MODELS APPLICATIONS

Given access to a learned score function, $\nabla_{x_t} \log p_\theta(x_t)$, from any pre-trained diffusion model, we denote the Jacobian of the score as $H_t(x_t)$. Assuming $M = 1$ in the OCM training objective, the covariance learning objective for diffusion models can be expressed as follows:

$$\min_\phi \frac{1}{T}\sum_{t=1}^{T} \mathbb{E}_{q(x_t,x_0)p(v)}||h_\phi(x_t) - v \odot H_t(x_t)v||_2^2, \tag{13}$$

where $v \sim p(v)$ and $h_\phi(x_t)$ is a network that conditioned on the state $x_t$ and time $t$. After training this objective, the learned $h_\phi(x_t)$ can be used to form the diagonal Hessian approximation $h_\phi(\tilde{x}) \approx \text{diag}(H(\tilde{x}))$ which further forms our approximation of covariance. We then derive its use cases in skip-step DDPM and DDIM for Diffusion acceleration.

**Skip-Step DDPM:** For the general skip-step DDPM with denoising distribution $q(x_{t'}|x_t)$ with $t' < t$. When $t' = t - 1$, this becomes the classic one-step DDPM. We further denote $\bar{\alpha}_{t':t} = \prod_{s=t'}^{t} \alpha_s$, and thus $\bar{\alpha}_{0:t} = \bar{\alpha}_t$, $\bar{\alpha}_{t':t} = \bar{\alpha}_t/\bar{\alpha}_{t'}$. We can write the forward process as

$$q(x_t|x_{t'}) = \mathcal{N}(x_t|\sqrt{\bar{\alpha}_{t':t}}x_{t'}, (1 - \bar{\alpha}_{t':t})I). \tag{14}$$

The corresponding Gaussian denoising distribution $p_{\theta,\phi}(x_{t'}|x_t) = \mathcal{N}(\mu_{t'}(x_t; \theta), \Sigma_{t'}(x_t; \phi))$ has the following mean and covariance functions:

$$\mu_{t'}(x_t; \theta) = (x_t + (1 - \bar{\alpha}_{t':t})\nabla_{x_t} \log p_\theta(x_t))/\sqrt{\bar{\alpha}_{t':t}}, \tag{15}$$

$$\Sigma_{t'}(x_t; \phi) = ((1 - \bar{\alpha}_{t':t})^2 h_\phi(x_t) + (1 - \bar{\alpha}_{t':t})I)/\bar{\alpha}_{t':t}. \tag{16}$$

Table 2: Overview of different covariance estimation methods. Methods are ranked by increasing modeling capability from top to bottom. We also include the intuition of the methods and how many additional network passes are required for estimating the covariance.

| Covariance Type | +#Passes | Intuition |
|---|---|---|
| $x_t$-independent Isotropic $\beta$ (Ho et al., 2020) | 0 | Cov. of $q(x_t\|x_{t-1})$ |
| $x_t$-independent Isotropic $\tilde{\beta}$ (Ho et al., 2020) | 0 | Cov. of $q(x_t\|x_{t-1}, x_0)$ |
| $x_t$-independent Isotropic Estimation (Bao et al., 2022b) | 0 | Estimate from data |
| $x_t$-dependent Diagonal VLB (Nichol & Dhariwal, 2021) | 1 | Learn from data |
| $x_t$-dependent Diagonal NS (Bao et al., 2022a) | 1 | Learn from data |
| $x_t$-dependent Diagonal OCM (Ours) | 1 | Learn from score |
| $x_t$-dependent Diagonal Estimation (Zhang et al., 2024b) | $2M$ | Estimate from score |
| $x_t$-dependent Diagonal Analytic (Zhang et al., 2024b) | $D$ | Calculate from score |
| $x_t$-dependent Full Analytic (Zhang et al., 2024b) | $D$ | Calculate from score |

*Modeling Capability* (vertical label with downward arrow on left side of table)

The skip-step denoising sample $x_{t'} \sim p_{\theta,\phi}(x_{t'}|x_t)$ can be obtained by $x_{t'} = \mu_{t'}(x_t;\theta) + \epsilon \Sigma_{t'}^{1/2}(x_t;\phi)$.

**Skip-Step DDIM:** Similarly, we give the skip-step formulation of DDIM, where we set the $\sigma_{t-1} = 0$ in Equation (6) which is also used in the original paper Song et al. (2020). We can use the approximated covariance of $p_{\theta,\phi}(x_0|x_t)$ to replace the delta function used in the vanilla DDIM, which gives the skip-steps DDIM sample

$$x_{t'} = \sqrt{\bar{\alpha}_{t'}} x_0 + \sqrt{1-\bar{\alpha}_{t'}}/\sqrt{1-\bar{\alpha}_t} \cdot (x_t - \sqrt{\bar{\alpha}_t} x_0), \qquad (17)$$

where $x_0 \sim p_{\theta,\phi}(x_0|x_t) = \mathcal{N}(\mu_0(x_t;\theta), \Sigma_0(x_t;\phi))$ and

$$\mu_0(x_t;\theta) = (x_t + (1-\bar{\alpha}_t)\nabla_{x_t}\log p_\theta(x_t))/\sqrt{\bar{\alpha}_t}, \qquad (18)$$

$$\Sigma_0(x_t;\phi) = ((1-\bar{\alpha}_t)^2 h_\phi(x_t,t) + (1-\bar{\alpha}_t)I)/\bar{\alpha}_t. \qquad (19)$$

The sample $x_0 \sim p_{\theta,\phi}(x_0|x_t)$ can be obtained by $x_0 = \mu_0(x_t;\theta) + \epsilon \Sigma_0^{1/2}(x_t;\phi)$. In the next section, we will compare the proposed method to other covariance estimation methods in practical examples.

### 3.3 DETAILS OF TRAINING AND INFERENCE

Our model comprises two components: a score prediction network $s_\theta$ and a diagonal Hessian prediction network $h_\phi$. In line with Bao et al. (2022a), we parameterize the score prediction network using a pretrained diffusion model, and the Hessian prediction network is parameterized by sharing parameters as follows:

$$s_\theta(x_t) = \mathrm{NN}_1(\mathrm{BaseNet}(x_t,t;\theta_1);\theta_2), \quad h_\phi(x_t) = \mathrm{NN}_2(\mathrm{BaseNet}(x_t,t;\theta_1);\phi) \qquad (20)$$

where BaseNet represents the commonly used architecture in diffusion models, such as UNet and DiT (Ho et al., 2020; Peebles & Xie, 2023). This parameterization approach only requires an additional small neural network, $\mathrm{NN}_2$, resulting in negligible extra computational and memory costs compared to the original diffusion models (see Appendix B.1 for more details). In our experiment, we fix the parameter $\theta = \{\theta_1, \theta_2\}$ and train the Hessian prediction network exclusively with the proposed OCM objective. After training, samples can be generated using Algorithms 1 and 2.

## 4 RELATED COVARIANCE ESTIMATION METHODS

We then discuss different choices of $\Sigma_{t'}(x_t)$ used in the diffusion model literature, see also Table 2 for an overview. For brevity, we mainly focus on DDPM here, in which the mean $\mu_{t'}(x_t)$ can be computed as in Equation (15).

**1. $x_t$-independent isotropic covariance: $\beta/\tilde{\beta}$-DDPM (Ho et al., 2020).** The $\beta$-DDPM uses the variance of $p(x_{t'}|x_t)$, which is $\Sigma_{t'}(x_t) = (1 - \bar{\alpha}_t/\bar{\alpha}_{t'})I$, when $t' = t-1$, we have $\Sigma_{t-1}(x_t) = \beta_t$. The $\tilde{\beta}$-DDPM uses the covariance of $p(x_{t'}|x_0, x_t)$, which is $\frac{(1-\bar{\alpha}_{t'})}{(1-\bar{\alpha}_t)}(1 - \bar{\alpha}_{t':t})$.

**2. $x_t$-independent isotropic covariance: A-DDPM (Bao et al., 2022b).** A-DDPM assumes a state-independent isotropic covariance $\Sigma_{t'}(x_t) = \sigma_{t'}^2 I$ with the following analytic form of

$$\sigma_{t'}^2 = \frac{1-\bar{\alpha}_{t':t}}{\bar{\alpha}_{t':t}} - \frac{(1-\bar{\alpha}_{t':t})^2}{d\bar{\alpha}_{t':t}}\mathbb{E}_{q(x_t)}\left[\|\nabla_{x_t}\log p_\theta(x_t)\|_2^2\right],$$

where the integration of $q(x_t)$ requires a Monte Carlo estimation before conducting generation. This variance choice is optimal under the KL divergence within a constrained isotropic state-independent posterior family.

**3. $x_t$-dependent diagonal covariance: I-DDPM (Nichol & Dhariwal, 2021).** In I-DDPM, the diagonal covariance matrix is modelled as the interpolation between $\beta_t$ and $\tilde{\beta}_t$

$$\Sigma_{t'}(x_t;\psi) = \exp(v_\psi(x_t)\log\beta_t + (1 - v_\psi(x_t))\log\tilde{\beta}_t), \tag{21}$$

where $v_\psi$ is parametrized via a neural network. The covariance is learned with the variational lower bound (VLB). In the optimal training, the covariance learned by VLB will recover the true covariance in (8). Notably, Nichol & Dhariwal (2021) heuristically obtain the covariance of skip sampling $\Sigma_{t'}(x_t;\psi)$ by rescaling $\beta_t$ and $\tilde{\beta}_t$ accordingly: $\beta_t \to 1 - \bar{\alpha}_{t':t}$, $\tilde{\beta}_t \to \frac{(1-\bar{\alpha}_{t'})}{(1-\bar{\alpha}_t)}(1 - \bar{\alpha}_{t':t})$. However, when $t' = 0$, $\Sigma_0(x_t;\psi)$ is ill-defined; thus, iDDPM is inapplicable within the DDIM framework.

**4. $x_t$-dependent diagonal covariance: SN-DDPM (Bao et al., 2022a).** SN-DDPM learns the covariance by training a neural network $g_\psi$ to estimate the second moment of the noise $\epsilon_t = (x_t - \sqrt{\bar{\alpha}_t}x_0)/\sqrt{1 - \bar{\alpha}_t}$:

$$\min_\psi \mathbb{E}_{t,q(x_0,x_t)}\|\epsilon_t^2 - g_\psi(x_t)\|_2^2. \tag{22}$$

After training, the covariance $\Sigma_{t'}(x_t;\psi)$ can be estimated via

$$\Sigma_{t'}(x_t;\psi) = \frac{(1-\bar{\alpha}_{t'})}{(1-\bar{\alpha}_t)}\bar{\beta}_{t':t}I + \frac{\beta_t^2}{\alpha_t}\left(\frac{g_\psi(x_t)}{1-\bar{\alpha}_t} - \nabla_{x_t}\log p_\theta(x_t)^2\right). \tag{23}$$

where $\bar{\beta}_{t':t} = 1 - \bar{\alpha}_{t':t}$. In optimal, $\Sigma_{t'}(x_t;\psi)$ in Equation (23) will recover the true covariance in Equation (8). We demonstrate the equivalence between OCM-DDPM and SN-DDPM in Appendix A.3. However, due to the appearance of the quadratic term in Equation (23), SN-DDPM tends to amplify the estimation error as $t \to 0$, leading to suboptimal solutions. To mitigate this issue, Bao et al. (2022a) also propose NPR-DDPM, which models the noise prediction residual instead. We recommend referring to their paper for detailed explanations.

Notably, almost all these methods can be applied within the DDIM framework by setting $t' = 0$. Specifically, $p(x_{t'}|x_t)$ can be sampled using Equation (17) with $x_0 \sim \mathcal{N}(\mu_0(x_t), \Sigma_0(x_t))$, where $\mu_0(x_t)$ is the same as in (18), but $\Sigma_0(x_t)$ differs for various methods as discussed previously.

## 5 EXPERIMENTAL RESULTS

To support our theoretical discussion, we first evaluate the performance of optimal covariance matching by training diffusion probabilistic models on 2D toy examples. We then demonstrate its effectiveness in enhancing image modelling in both pixel and latent spaces, focusing on the comparison between optimal covariance matching and other covariance estimation methods, and showing that the proposed approach has the potential to scale to large image generation tasks.

### 5.1 TOY DEMONSTRATION

We first demonstrate the effectiveness of our method by considering the data distribution as a two-dimensional mixture of forty Gaussians (MoG) with means uniformly distributed over $[-40, 40] \otimes [-40, 40]$ and a standard deviation of $\sigma = \sqrt{40}$, where $\otimes$ denotes Cartesian product (see Figure 1a for visualization). In this case, both the true score $\nabla_{x_t}\log p(x_t)$ and the optimal covariance $\Sigma_{t'}(x_t)$ are available, allowing us to compare the covariance estimation error. We then learn the covariance using the true scores by different methods to conduct DDPM and DDIM sampling for this MoG

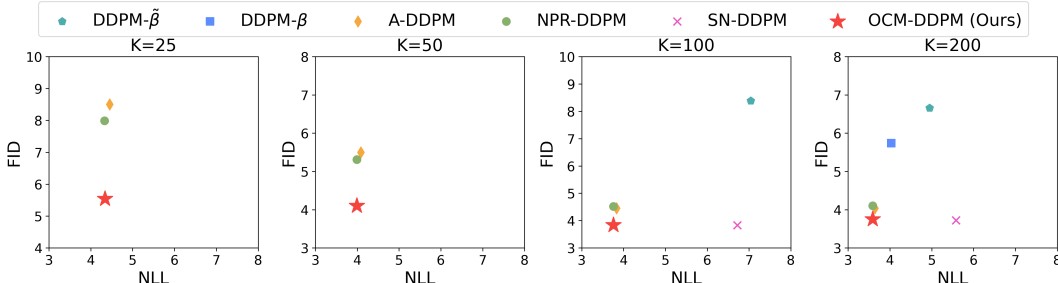

Figure 2: The results of FID v.s. NLL for different methods with varying numbers of sampling steps on CIFAR10 (CS). Our method consistently achieves the best trade-off between FID and NLL.

problem. For evaluation, we employ the Maximum Mean Discrepancy (MMD) (Gretton et al., 2012), which utilizes five kernels with bandwidths $\{2^{-2}, 2^{-1}, 2^0, 2^1, 2^2\}$. In Figures 4c and 4d, we show the MMD comparison among different methods. Specifically, we choose to conduct different diffusion steps with the skip-step scheme as discussed in Section 3.2.

The results, shown in Figures 1b and 1c, demonstrate that the proposed method outperforms other covariance learning approaches. Additionally, we include two methods utilizing the true diagonal and full covariance, serving as benchmarks for the best achievable performance. Notably, in this case, using the full covariance yields better performance compared to the diagonal covariance. This ob-

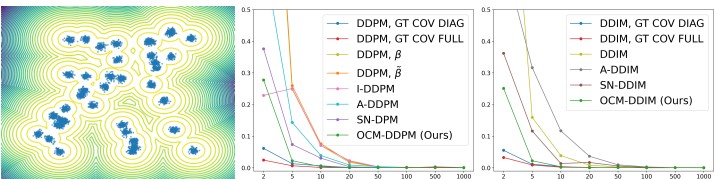

(a) Training Data and Density  (b) DDPM MMD v.s. Steps  (c) DDIM MMD v.s. Steps

Figure 1: Comparisons of different covariance estimation methods. Figure (a) demonstrates the training data and the ground truth density. Figures (b) and (c) present the MMD evaluation against the total sampling steps in the DDPM (b) and DDIM (c) settings.

servation highlights the importance of accurate covariance approximation, suggesting that improved methods for learning the off-diagonal terms could lead to even better results. For further analysis, we present an additional toy experiment in Appendix C.1 to demonstrate the efficacy of our approach.

## 5.2 IMAGE MODELLING WITH DIFFUSION MODELS

Following the experimental setting in Bao et al. (2022a), we then evaluate our method across varying pre-trained score networks provided by previous works (Ho et al., 2020; Nichol & Dhariwal, 2021; Song et al., 2020; Bao et al., 2022b; Peebles & Xie, 2023). In this experiment, we mainly focus on four datasets: CIFAR10 (Krizhevsky et al., 2009) with the linear schedule (LS) of $\beta_t$ (Ho et al., 2020) and the cosine schedule (CS) of $\beta_t$ (Nichol & Dhariwal, 2021); CelebA (Liu et al., 2015); LSUN Bedroom (Yu et al., 2015). The details of the experimental setting can be found in Appendix B.

### IMPROVING LIKELIHOOD RESULTS

We first evaluate the negative log-likelihood (NLL) of the diffusion models by calculating the negative evidence lower bound (ELBO): $-\log q(x_0) \leq \mathbb{E}_q \log \frac{p(x_{0:T})}{q(x_{1:T}|x_0)} \equiv -L_{\text{elbo}}(x_0)$ with the same mean yet varying covariances. As per Bao et al. (2022b), we report results only for the DDPM forward process, as $-L_{\text{elbo}} = \infty$ under the DDIM forward process. As shown in Table 3, our OCM-DDPM demonstrates the best or second-best performance in most scenarios, highlighting the effectiveness of our covariance estimation approach. We also note that SN-DDPM performs poorly on likelihood results in small-scale datasets like CIFAR10 and CELEBA, likely due to the amplified error of the quadratic term in Equation (22). Although NPR-DDPM achieves slightly better performance in certain scenarios, it falls short in terms of sample quality, as measured by FID, shown in Table 5. In Appendix C.2, we also compare our method to Improved DDPM (Nichol & Dhariwal, 2021) in terms of likelihood on ImageNet 64x64. The results show that our method achieves better likelihood estimation with a small number of timesteps, while achieving comparable results with full timesteps.

Table 3: The NLL (bits/dim) ↓ across various datasets using different sampling steps.

| | CIFAR10 (LS) | | | | | | CIFAR10 (CS) | | | | | |
|---|---|---|---|---|---|---|---|---|---|---|---|---|
| # TIMESTEPS $K$ | 10 | 25 | 50 | 100 | 200 | 1000 | 10 | 25 | 50 | 100 | 200 | 1000 |
| DDPM, $\tilde{\beta}$ | 74.95 | 24.98 | 12.01 | 7.08 | 5.03 | 3.73 | 75.96 | 24.94 | 11.96 | 7.04 | 4.95 | 3.60 |
| DDPM, $\beta$ | 6.99 | 6.11 | 5.44 | 4.86 | 4.39 | 3.75 | 6.51 | 5.55 | 4.92 | 4.41 | 4.03 | 3.54 |
| A-DDPM | 5.47 | 4.79 | 4.38 | 4.07 | 3.84 | 3.59 | 5.08 | 4.45 | 4.09 | 3.83 | 3.64 | 3.42 |
| NPR-DDPM | 5.40 | 4.64 | 4.25 | 3.98 | 3.79 | 3.57 | 5.03 | 4.33 | 3.99 | 3.76 | 3.59 | 3.41 |
| SN-DDPM | 30.79 | 11.83 | 7.13 | 5.24 | 4.39 | 3.74 | 90.85 | 19.81 | 9.72 | 6.72 | 5.58 | 4.73 |
| OCM-DDPM | 5.32 | 4.63 | 4.25 | 3.97 | 3.78 | 3.57 | 4.99 | 4.34 | 3.99 | 3.76 | 3.59 | 3.41 |

| | CELEBA 64x64 | | | | | | IMAGENET 64x64 | | | | | |
|---|---|---|---|---|---|---|---|---|---|---|---|---|
| # TIMESTEPS $K$ | 10 | 25 | 50 | 100 | 200 | 1000 | 25 | 50 | 100 | 200 | 400 | 4000 |
| DDPM, $\tilde{\beta}$ | 33.42 | 13.09 | 7.14 | 4.60 | 3.45 | 2.71 | 105.87 | 46.25 | 22.02 | 12.10 | 7.59 | 3.89 |
| DDPM, $\beta$ | 6.67 | 5.72 | 4.98 | 4.31 | 3.74 | 2.93 | 5.81 | 5.20 | 4.70 | 4.31 | 4.04 | 3.65 |
| A-DDPM | 4.54 | 3.89 | 3.48 | 3.16 | 2.92 | 2.66 | 4.78 | 4.42 | 4.15 | 3.95 | 3.81 | 3.61 |
| NPR-DDPM | 4.46 | 3.78 | 3.40 | 3.11 | 2.89 | 2.65 | 4.66 | 4.22 | 3.96 | 3.80 | 3.71 | 3.60 |
| SN-DDPM | 18.09 | 8.05 | 5.29 | 4.05 | 3.40 | 2.84 | 4.56 | 4.18 | 3.95 | 3.80 | 3.71 | 3.63 |
| OCM-DDPM | 4.69 | 3.86 | 3.43 | 3.13 | 2.90 | 2.66 | 4.45 | 4.15 | 3.93 | 3.79 | 3.70 | 3.59 |

## IMPROVING SAMPLE QUALITY

Next, we compare sample quality quantitatively using the FID score, with results reported for both the DDPM and DDIM forward processes. As shown in Table 5, our OCM-DPM consistently achieves either the best or second-best results across most settings. The qualitative results of the samples generated by our method can be found in Appendix C.6.

Notably, while SN-DDPM often demonstrates slightly better FID performance in certain cases, it struggles with likelihood estimation, as shown in Table 3. Conversely, as discussed in the previous section, NPR-DDPM sometimes achieves superior likelihood results in Table 3, but its image generation quality is significantly worse than OCM-DDPM. Therefore, the proposed OCM-DDPM provides a more balanced model, offering a better trade-off between generation quality and likelihood estimation. To better highlight this benefit, we include a visualization in Figure 2.

## 5.3 IMAGE MODELLING WITH LATENT DIFFUSION MODELS

In this section, we apply our methods to latent diffusion models (Vahdat et al., 2021; Rombach et al., 2022) and with ImageNet 256×256 to demonstrate the scalability of our method. Specifically, we compare our methods to other approaches within the DiT architecture (Peebles & Xie, 2023), evaluating sample quality using the FID score and diversity using the Recall score (Sajjadi et al., 2018). We focus on conditional generation with classifier-free guidance (CFG), which is consistent to Peebles & Xie (2023).

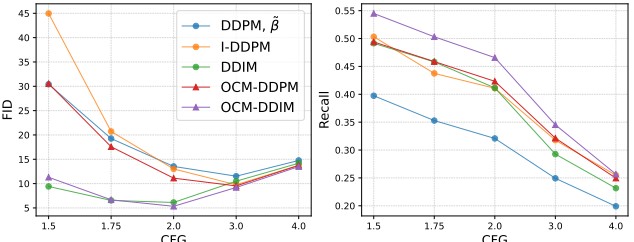

Figure 3: Results of DiT training on ImageNet 256x256. We generate samples using 10 timesteps with varying CFG coefficients (see Table 12 for exact numerical values).

It is important to note that DiT was initially trained using the Improved DDPM (I-DDPM) algorithm (Nichol & Dhariwal, 2021). Therefore, the results of I-DDPM reflect the original performance of the pre-trained DiT model[1] provided by Peebles & Xie (2023). For DDPM and DDIM sampling, we discard the learned covariance in DiT and use only the learned noise prediction neural network for the posterior mean. In contrast, our OCM-DPM retains the same mean function while learning the covariance using the optimal covariance matching objective. We then generate samples with 10 sampling steps and report FID and Recall scores across different CFG coefficients on the ImageNet 256x256 dataset. The results are displayed in Figure 3, with DDPM-$\beta$ excluded due to its poor performance. It shows that our OCM-DPM method achieves the best FID performance with CFG = 2.0. Although DDPM-$\tilde{\beta}$ and DDIM perform better in terms of FID at CFG = 1.5, their Recall scores

---

[1]We use the DiT XL/2 checkpoint released in https://github.com/facebookresearch/DiT.

Table 4: The least number of timesteps ↓ required to achieve an FID around 6 (along with the corresponding FID). To account for the additional time cost incurred by the covariance prediction network, we multiply the results by the ratio of the time cost per single timestep, reflecting the extra computational overhead (see Appendix B.1 for details).

| METHOD | CIFAR10 | CELEBA 64x64 | LSUN BEDROOM | IMAGENET 256x256 |
|---|---|---|---|---|
| DDPM | 90 (6.12) | $> 200$ | 130 (6.06) | 21 (5.89) |
| DDIM | 30 (5.85) | $> 100$ | BEST FID $> 6$ | 11 (5.58) |
| IMPROVED DDPM | 45 (5.96) | MISSING MODEL | **90** (6.02) | 22 (6.08) |
| ANALYTIC-DPM | 25 (5.81) | 55 (5.98) | 100 (6.05) | MISSING MODEL |
| NPR-DPM | $1.002 \times 23$ (5.76) | $1.013 \times 50$ (6.04) | $1.021 \times$**90** (6.01) | MISSING MODEL |
| SN-DPM | $1.005 \times 17$ (5.81) | $1.019 \times 22$ (5.96) | $1.114 \times 92$ (6.02) | MISSING MODEL |
| OCM-DPM (OURS) | $1.003 \times$**16** (5.83) | $1.015 \times$**21** (5.94) | $1.112 \times$**90** (6.04) | $1.007 \times$**10** (5.33) |

are lower than those of the proposed OCM methods. This discrepancy arises partly because DDPM and DDIM model the inverse process with deterministic variance, whereas OCM methods explicitly learn the variance from data, allowing for more accurate density estimation and generating more diverse results. As Peebles & Xie (2023) report that DiT achieves the best FID performance with CFG = 1.5, we further evaluate performance with different sampling steps under CFG = 1.5, as shown in Table 13. The results indicate that our OCM-DPM methods offer a great improvement in sample quality and diversity.

To provide an overview of the superiority of our method, we compare the minimum sampling step to achieve the FID score (Heusel et al., 2017) close to 6 across different approaches (see Appendix B.1 for details). As shown in Table 4, our method requires the fewest denoising steps in most settings. Moreover, we emphasize that compared to recent advanced methods like SN-DPM and NPR-DPM, *our approach is the only one that consistently delivers both competitive likelihood and FID*.

## 6 RELATED WORK AND FUTURE DIRECTIONS

In this paper, we show that improving diagonal covariance estimation can significantly enhance the performance of diffusion models. This raises a natural question: can more flexible covariance structures further improve these models? In Figure 1, we demonstrate that a full covariance structure achieves better generation quality with fewer NFEs in a toy 2D problem. However, for high-dimensional problems, the quadratic growth in parameter scale makes a full covariance approach computationally impractical. To address this, low-rank or block-diagonal covariance approximations offer promising alternatives. Developing effective training objectives for flexible covariance structures remains a compelling direction for future research.

In addition to the covariance estimation methods discussed in Section 4, there are other approaches to accelerate sampling in diffusion models. One approach involves using faster numerical solvers for differential equations with continuous timesteps (Jolicoeur-Martineau et al., 2021; Liu et al., 2022; Lu et al., 2022). Another strategy, inspired by Schrödinger bridge (Wang et al., 2021; De Bortoli et al., 2021), is to introduce a nonlinear, trainable forward diffusion process. Additionally, replacing Gaussian modelling of $p_\theta(x_0|x_t)$ with more expressive alternatives, such as GANs (Xiao et al., 2021), distributional models (Bortoli et al., 2025), latent variable models (Yu et al., 2024), or energy-based models (Xu et al., 2024), can also accelerate the sampling of diffusion models.

Recently, distillation techniques have gained popularity, achieving state-of-the-art in one-step generation (Zhou et al., 2024). There are two prominent types of distillation methods. The first is trajectory distillation (Salimans & Ho, 2022; Berthelot et al., 2023; Song et al., 2023; Heek et al., 2024; Kim et al., 2023; Li & He, 2024), which focuses on accelerating the process of solving differential equations. The second type involves distillation techniques that utilize a one-step implicit latent variable model as the student model (Luo et al., 2024; Salimans et al., 2024; Xie et al., 2024; Zhou et al., 2024; Zhang et al., 2025) and distills the diffusion process into the student model by minimizing the spread divergence family (Zhang et al., 2020; 2019) through score estimation (Poole et al., 2022; Wang et al., 2024).

Although these distillation methods typically offer faster generation speeds (fewer NFEs) compared to covariance estimation methods, they often lack tractable density or likelihood estimation. This presents a challenge for generative modeling applications where likelihood or density estimation is

Table 5: FID score ↓ across various datasets using different sampling steps.

| # TIMESTEPS $K$ | CIFAR10 (LS) | | | | | | CIFAR10 (CS) | | | | | |
|---|---|---|---|---|---|---|---|---|---|---|---|---|
| | 10 | 25 | 50 | 100 | 200 | 1000 | 10 | 25 | 50 | 100 | 200 | 1000 |
| DDPM, $\tilde\beta$ | 44.45 | 21.83 | 15.21 | 10.94 | 8.23 | 5.11 | 34.76 | 16.18 | 11.11 | 8.38 | 6.66 | 4.92 |
| DDPM, $\beta$ | 233.41 | 125.05 | 66.28 | 31.36 | 12.96 | **3.04** | 205.31 | 84.71 | 37.35 | 14.81 | 5.74 | **3.34** |
| A-DDPM | 34.26 | 11.60 | 7.25 | 5.40 | 4.01 | 4.03 | 22.94 | 8.50 | 5.50 | 4.45 | 4.04 | 4.31 |
| NPR-DDPM | 32.35 | 10.55 | 6.18 | 4.52 | 3.57 | 4.10 | 19.94 | 7.99 | 5.31 | 4.52 | 4.10 | 4.27 |
| SN-DDPM | **24.06** | **6.91** | **4.63** | **3.67** | **3.31** | 3.65 | 16.33 | 6.05 | 4.17 | **3.83** | **3.72** | 4.07 |
| OCM-DDPM | 24.94 | 9.19 | 5.95 | 4.36 | 3.48 | 3.98 | **14.32** | **5.54** | 4.10 | 3.84 | 3.75 | 4.18 |
| DDIM | 21.31 | 10.70 | 7.74 | 6.08 | 5.07 | 4.13 | 34.34 | 16.68 | 10.48 | 7.94 | 6.69 | 4.89 |
| A-DDIM | 14.00 | 5.81 | 4.04 | 3.55 | 3.39 | 3.74 | 26.43 | 9.96 | 6.02 | 4.88 | 4.92 | 4.66 |
| NPR-DDIM | 13.34 | 5.38 | 3.95 | 3.53 | 3.42 | 3.72 | 22.81 | 9.47 | 6.04 | 5.02 | 5.06 | 4.62 |
| SN-DDIM | 12.19 | **4.28** | **3.39** | 3.23 | 3.22 | 3.65 | 17.90 | 7.36 | 5.16 | 4.63 | 4.63 | 4.51 |
| OCM-DDIM | **10.66** | 4.35 | 3.48 | 3.27 | 3.29 | 3.74 | **16.70** | **6.71** | 4.72 | **4.30** | 4.54 | 4.53 |

| # TIMESTEPS $K$ | CELEBA 64x64 | | | | | | IMAGENET 64x64 | | | | | |
|---|---|---|---|---|---|---|---|---|---|---|---|---|
| | 10 | 25 | 50 | 100 | 200 | 1000 | 25 | 50 | 100 | 200 | 400 | 4000 |
| DDPM, $\tilde\beta$ | 36.69 | 24.46 | 18.96 | 14.31 | 10.48 | 5.95 | 29.21 | 21.71 | 19.12 | 17.81 | 17.48 | 16.55 |
| DDPM, $\beta$ | 294.79 | 115.69 | 53.39 | 25.65 | 9.72 | **3.16** | 170.28 | 83.86 | 45.04 | 28.39 | 21.38 | 16.38 |
| A-DDPM | 28.99 | 16.01 | 11.23 | 8.08 | 6.51 | 5.21 | 32.56 | 22.45 | 18.80 | 17.16 | 16.40 | 16.34 |
| NPR-DDPM | 28.37 | 15.74 | 10.89 | 8.23 | 7.03 | 5.33 | 28.27 | 20.89 | 18.06 | 16.96 | **16.32** | 16.38 |
| SN-DDPM | **20.60** | **12.00** | **7.88** | **5.89** | **5.02** | 4.42 | **27.58** | **20.74** | 18.04 | **16.61** | 16.37 | **16.22** |
| OCM-DDPM | 21.55 | 12.71 | 9.24 | 6.97 | 5.92 | 5.04 | 28.02 | 20.81 | **17.98** | 16.74 | **16.32** | 16.31 |
| DDIM | 20.54 | 13.45 | 9.33 | 6.60 | 4.96 | 3.40 | 26.06 | 20.10 | 18.09 | 17.84 | 17.74 | 19.00 |
| A-DDIM | 15.62 | 9.22 | 6.13 | 4.29 | 3.46 | 3.13 | **25.98** | **19.23** | 17.73 | 17.49 | 17.44 | 18.98 |
| NPR-DDIM | 14.98 | 8.93 | 6.04 | 4.27 | 3.59 | 3.15 | 28.84 | 19.62 | 17.63 | 17.42 | 17.30 | 18.91 |
| SN-DDIM | **10.20** | **5.48** | **3.83** | 3.04 | 2.85 | 2.90 | 28.07 | 19.38 | 17.53 | 17.23 | 17.23 | **18.89** |
| OCM-DDIM | 10.28 | 5.72 | 4.42 | 3.54 | 3.17 | 3.03 | 28.28 | 19.62 | 17.71 | 17.42 | 17.26 | 19.02 |

crucial. For example, in AI4Science problems involving energy-based data, accurate model density estimation is essential for enabling importance sampling to correct sample bias, as discussed in (Zhang et al., 2024a; Vargas et al., 2023; Chen et al., 2024; Akhound-Sadegh et al., 2024). Another example is diffusion model-based data compression, where better likelihood corresponds to better compression rates (Townsend et al., 2019; Zhang et al., 2021; Ho et al., 2020; Kingma et al., 2021). In these cases, the proposed OCM-DDPM can be used to provide better likelihood estimates, resulting in improved task performance. Additionally, a recent paper (Salimans et al., 2024) shows how moment-matching improves distillation, achieving state-of-the-art diffusion quality with first-order matching. Our method could extend this to higher-order moment matching, potentially accelerating distillation training. We leave it as a promising direction for future work.

Beyond image modelling, our method can be straightforwardly applied to accelerate large-scale video diffusion models (Blattmann et al., 2023; Chen et al., 2023), which are based on latent diffusion models. A recent study (Zhao et al., 2024) shows that covariance estimation is crucial in mitigating the image-leakage issue in image-to-video generation problems, presenting another promising application of our method. Moreover, our method can be applied to solve inverse problems, where the optimal covariance can enhance the accuracy of posterior sampling (Chung et al., 2022; Rozet et al., 2024). While this work focuses on the DDPM and DDIM sampler, incorporating the improved covariance into the general stochastic and deterministic differential solver (Song et al., 2021; Karras et al., 2022) also represents exciting directions for future research.

## 7 CONCLUSION

In this paper, we proposed a new method for learning the diagonal covariance of the denoising distribution, which offers improved accuracy compared to other covariance learning approaches. Our results demonstrate that enhanced covariance estimation leads to better generation quality and diversity, all while reducing the number of required generation steps. We validated the scalability of our method across different problem domains and scales and discussed its connection to various acceleration techniques and highlighted several promising application areas for future exploration.

## 8  ACKNOWLEDGMENTS

ZO is supported by the Lee Family Scholarship. MZ and DB acknowledge funding from AI Hub in Generative Models, under grant EP/Y028805/1 and funding from the Cisco Centre of Excellence. We want to thank Wenlin Chen for the useful discussions.

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

# Appendix for "Improved Diffusion Models with Optimal Covariance Matching"

## CONTENTS

## A  ABSTRACT PROOF AND DERIVATIONS

### A.1  DERIVATIONS OF THE ANALYTICAL FULL COVARIANCE IDENTITY

**Theorem 1** (Generalized Analytical Covariance Identity). *Given a joint distribution $q(\tilde{x}, x) = q(\tilde{x}|x)q(x)$ with $q(\tilde{x}|x) = \mathcal{N}(\alpha x, \sigma^2 I)$, then the covariance of the true posterior $q(x|\tilde{x}) \propto q(x)q(\tilde{x}|x)$, which is defined as $\Sigma(\tilde{x}) = \mathbb{E}_{q(x|\tilde{x})}[x^2] - \mathbb{E}_{q(x|\tilde{x})}[x]^2$, has a closed form:*

$$\Sigma(\tilde{x}) = \left(\sigma^4 \nabla_{\tilde{x}}^2 \log q(\tilde{x}) + \sigma^2 I\right)/\alpha^2. \tag{8}$$

*Proof.* This proof generalizes the original analytical covariance identity proof discussed in Zhang et al. (2024b). Using the fact that $\nabla_{\tilde{x}} q(\tilde{x}|x) = -\frac{1}{\sigma^2}(\tilde{x} - \alpha x)q(\tilde{x}|x)$ and the Tweedie's formula $\nabla_{\tilde{x}} \log q(\tilde{x}) = \frac{1}{\sigma^2}\left(\alpha \mathbb{E}_{q(x|\tilde{x})}[x] - \tilde{x}\right)$, we can expand the hessian of the $\log q_\theta(\tilde{x})$:

$$\nabla_{\tilde{x}}^2 \log q(\tilde{x}) = -\frac{1}{\sigma^2} \int \nabla_{\tilde{x}}\left((\tilde{x} - \alpha x)\frac{q(\tilde{x}|x)q(x)}{q(\tilde{x})}\right) \mathrm{d}x$$

$$= -\frac{1}{\sigma^2} \int \frac{q(\tilde{x}|x)q(x)}{q(\tilde{x})} \mathrm{d}x - \frac{1}{\sigma^2} \int (\tilde{x} - \alpha x)\frac{\nabla_{\tilde{x}}q(\tilde{x}|x)q(\tilde{x})q(x) - \nabla_{\tilde{x}}q(\tilde{x})q(\tilde{x}|x)q(x)}{q^2(\tilde{x})} \mathrm{d}x$$

$$\implies \sigma^2 \nabla_{\tilde{x}}^2 \log q(\tilde{x}) + 1 = -\int (\tilde{x} - \alpha x)\frac{\nabla_{\tilde{x}}q(\tilde{x}|x)q(x) - \nabla_{\tilde{x}}\log q(\tilde{x})q(\tilde{x}|x)q(x)}{q(\tilde{x})} \mathrm{d}x$$

$$= -\int (\tilde{x} - \alpha x)\frac{-\frac{1}{\sigma^2}(\tilde{x} - \alpha x)q(\tilde{x}|x)q(x) + \frac{1}{\sigma^2}(\tilde{x} - \alpha\mathbb{E}_{q(x|\tilde{x})}[x])q(\tilde{x}|x)q(x)}{q(\tilde{x})} \mathrm{d}x$$

$$\implies \sigma^4 \nabla_{\tilde{x}}^2 \log q(\tilde{x}) + \sigma^2 I = \int \left((\tilde{x} - \alpha x)^2 - (\tilde{x} - \alpha x)(\tilde{x} - \alpha\mathbb{E}_{q(x|\tilde{x})}[x])\right) q(x|\tilde{x}) \mathrm{d}x$$

$$= \alpha^2 \mathbb{E}_{q(x|\tilde{x})}[x^2] - \alpha^2 \mathbb{E}_{q(x|\tilde{x})}[x]^2 \equiv \alpha^2 \Sigma(\tilde{x})$$

Therefore, we obtain the analytical full covariance identity: $\Sigma_q(\tilde{x}) = \left(\sigma^4 \nabla_{\tilde{x}}^2 \log q(\tilde{x}) + \sigma^2 I\right)/\alpha^2$.

$\square$

## A.2 VALIDITY OF THE OCM OBJECTIVE

**Theorem 2** (Validity of the OCM objective). *The objective in Equation* (11) *upper bounded the base objective (i.e., Equation* (10) *with* $M \to \infty$). *Moreover, it attains optimal when* $h_\phi(\tilde{x}) = diag(H(\tilde{x}))$ *for all* $\tilde{x} \sim q(\tilde{x})$.

*Proof.* Recall that to learn the optimal covariance, we can minimize the following loss function

$$\min_\phi \mathbb{E}_{q(\tilde{x})} \left\| h_\phi(\tilde{x}) - \mathbb{E}_{q(v)}[v \odot H(\tilde{x})v] \right\|_2^2 \tag{24}$$

We call this the grounded objective because it remains unbiased and consistent, i.e., $h_{\phi^*} = diag(H(\tilde{x}))$, when the inner expectation is approximated with infinite Monte Carlo samples. Using Jensen's inequality, we can show that the OCM objective defined in (11) provides an upper bound for the grounded objective

$$\mathbb{E}_{q(\tilde{x})} \left\| h_\phi(\tilde{x}) - \mathbb{E}_{q(v)}[v \odot H(\tilde{x})v] \right\|_2^2 = \mathbb{E}_{q(\tilde{x})} \left\| \mathbb{E}_{q(v)}[h_\phi(\tilde{x}) - v \odot H(\tilde{x})v] \right\|_2^2$$
$$\leq \mathbb{E}_{q(\tilde{x})} \mathbb{E}_{q(v)} \left\| h_\phi(\tilde{x}) - v \odot H(\tilde{x})v \right\|_2^2$$
$$= L_{ocm}(\phi).$$

Thus, minimizing the OCM objective also minimizes the grounded objective, leading to a more accurate approximation of the diagonal Hessian. We then show that these two objectives are equivalent when attaining their optimal. To see this, we can expand the OCM objective

$$L_{ocm}(\phi) = \mathbb{E}_{p(v)q(\tilde{x})} \|h_\phi(\tilde{x}) - v \odot H(\tilde{x})v\|_2^2$$
$$= \mathbb{E}_{q(\tilde{x})} \|h_\phi(\tilde{x})\|_2^2 - 2\mathbb{E}_{p(v)q(\tilde{x})}[h_\phi(\tilde{x})^T(v \odot H(\tilde{x})v)] + c$$
$$= \mathbb{E}_{q(\tilde{x})} \|h_\phi(\tilde{x})\|_2^2 - 2\mathbb{E}_{q(\tilde{x})}[h_\phi(\tilde{x})^T diag(H(\tilde{x}))] + c$$
$$= \mathbb{E}_{q(\tilde{x})} \|h_\phi(\tilde{x}) - diag(H(\tilde{x}))\|_2^2 + c',$$

where $c, c'$ are constants w.r.t. the parameter $\phi$, and line 2 to line 3 follows from the fact that $\mathbb{E}_{p(v)}[v \odot H(x)v] = diag(H(x))$. Therefore, $L_{ocm}(\phi)$ attains optimal when $h_\phi(\tilde{x}) = diag(H(\tilde{x}))$ for all $\tilde{x} \sim q(\tilde{x})$. $\square$

## A.3 CONNECTION TO SN-DDPM

In this section, we showcase the connection between OCM-DDPM and SN-DDPM through the second-order Tweedie's formula. Before delving into it, we present the following lemmas, which are essential for our derivation.

**Lemma 1** (First order Tweedie's formula (Efron, 2011)). *Let* $q(\tilde{x}|x) = \mathcal{N}(\tilde{x}|\sqrt{\alpha}x, \beta I)$, *we have the mean of the inverse density equals*

$$\mathbb{E}_{q(x|\tilde{x})}[x] = \frac{1}{\sqrt{\alpha}}(\tilde{x} + \beta \nabla_{\tilde{x}} \log q(\tilde{x})). \tag{25}$$

**Lemma 2** (Second order Tweedie's formula). *Let* $q(\tilde{x}|x) = \mathcal{N}(\tilde{x}|\sqrt{\alpha}x, \beta I)$, *we have the second moment of the inverse density equals*

$$\mathbb{E}_{q(x|\tilde{x})}[xx^T] = \frac{1}{\alpha} \left( \tilde{x}\tilde{x}^T + \beta s_1(\tilde{x})\tilde{x}^T + \beta \tilde{x} s_1(\tilde{x})^T + \beta^2 s_2(\tilde{x}) + \beta^2 s_1(\tilde{x})s_1(\tilde{x})^T + \beta I \right), \tag{26}$$

*where* $s_1(\tilde{x}) \equiv \nabla_{\tilde{x}} \log q(\tilde{x})$ *and* $s_2(\tilde{x}) \equiv \nabla_{\tilde{x}}^2 \log q(\tilde{x})$.

*Proof.* The proof follows that in (Meng et al., 2021, Appendix B) with the generalization to the scaled Gaussian convolutions. Specifically, we first reparametrized $q(\tilde{x}|x)$ as a exponential distribution

$$q(\tilde{x}|\eta) = e^{\eta^T \tilde{x} - \psi(\eta)} q_0(\tilde{x}), \tag{27}$$

where $\eta = \frac{\sqrt{\alpha}}{\beta}x$, $q_0(\tilde{x}) \propto e^{-\frac{1}{2\beta}\tilde{x}^T\tilde{x}}$, and $\psi(\eta)$ denotes the partition function. By applying the Bayes rule $q(\eta|\tilde{x}) = \frac{q(\tilde{x}|\eta)p(\eta)}{q(\tilde{x})}$, we have the corresponding posterior

$$q(\eta|\tilde{x}) = e^{\eta^T \tilde{x} - \psi(\eta) - \lambda(\tilde{x})} p(\eta), \tag{28}$$

where $\lambda(\tilde{x}) \equiv \log q(\tilde{x}) - \log q_0(\tilde{x})$. Since $\int q(\eta|\tilde{x})\mathrm{d}\eta = 1$, by taking the derivative w.r.t. $\tilde{x}$ on both sides, we have

$$\int (\eta - \nabla_{\tilde{x}}\lambda(\tilde{x}))^T q(\eta|\tilde{x}) = 0, \tag{29}$$

which implies that $\mathbb{E}[\eta|\tilde{x}] = \nabla_{\tilde{x}}\lambda(\tilde{x})$. Taking the derivative w.r.t. $\tilde{x}$ on both sides again, we have

$$\int \eta (\eta - \nabla_{\tilde{x}}\lambda(\tilde{x}))^T q(\eta|\tilde{x}) = \nabla_{\tilde{x}}^2\lambda(\tilde{x}), \tag{30}$$

which implies that $\mathbb{E}[\eta\eta^T|\tilde{x}] = \nabla_{\tilde{x}}^2\lambda(\tilde{x}) + \nabla_{\tilde{x}}\lambda(\tilde{x})\nabla_{\tilde{x}}^T\lambda(\tilde{x})$. By substituting $\eta = \frac{\sqrt{\alpha}}{\beta}x$, $\nabla_{\tilde{x}}\lambda(\tilde{x}) = s_1(\tilde{x}) + \frac{1}{\beta}\tilde{x}$, and $\nabla_{\tilde{x}}^2\lambda(\tilde{x}) = s_2(\tilde{x}) + \frac{1}{\beta}I$, we get the result as desired. $\square$

**Lemma 3** (Convert the covariance of $q(\tilde{x}|x)$ to the hessian of $q(\tilde{x})$). *Let $q(\tilde{x}|x) = \mathcal{N}(\tilde{x}|\sqrt{\alpha}x, \beta I)$, we have the covariance of the inverse density equals*

$$\mathrm{Cov}_{q(x|\tilde{x})}[x] = \frac{\beta}{\alpha}(I + \beta\nabla_{\tilde{x}}^2 \log q(\tilde{x})). \tag{31}$$

*Proof.* Let $s_1(\tilde{x}) \equiv \nabla_{\tilde{x}} \log q(\tilde{x})$ and $s_2(\tilde{x}) \equiv \nabla_{\tilde{x}}^2 \log q(\tilde{x})$. We have

$$\mathrm{Cov}_{q(x|\tilde{x})}[x] = \frac{\beta^2}{\alpha}\mathrm{Cov}_{q(x|\tilde{x})}\left[\frac{\tilde{x} - \sqrt{\alpha}x}{\beta}\right]$$

$$= \frac{\beta^2}{\alpha}\left(\mathbb{E}_{q(x|\tilde{x})}\left(\frac{\tilde{x}-\sqrt{\alpha}x}{\beta}\right)\left(\frac{\tilde{x}-\sqrt{\alpha}x}{\beta}\right)^T - \mathbb{E}_{q(x|\tilde{x})}\left(\frac{\tilde{x}-\sqrt{\alpha}x}{\beta}\right)\mathbb{E}_{q(x|\tilde{x})}\left(\frac{\tilde{x}-\sqrt{\alpha}x}{\beta}\right)^T\right)$$

$$= \frac{\beta^2}{\alpha}\left(\frac{1}{\beta^2}\mathbb{E}_{q(x|\tilde{x})}\left(\tilde{x} - \sqrt{\alpha}x\right)\left(\tilde{x} - \sqrt{\alpha}x\right)^T - s_1(\tilde{x})s_1(\tilde{x})^T\right)$$

$$= \frac{\beta^2}{\alpha}\left(\frac{1}{\beta^2}\left(\tilde{x}\tilde{x}^T - 2\tilde{x}(\tilde{x} + \beta s_1(\tilde{x}))^T + \alpha\mathbb{E}_{q(x|\tilde{x})}[xx^T]\right) - s_1(\tilde{x})s_1(\tilde{x})^T\right)$$

$$= \frac{\beta^2}{\alpha}\left(\frac{1}{\beta^2}\left(\beta^2 s_2(\tilde{x}) + \beta^2 s_1(\tilde{x})s_1(\tilde{x})^T + \beta I\right) - s_1(\tilde{x})s_1(\tilde{x})^T\right)$$

$$= \frac{\beta^2}{\alpha}\left(\frac{1}{\beta}I + s_2(\tilde{x})\right) \equiv \frac{\beta}{\alpha}\left(I + \beta\nabla_{\tilde{x}}^2 \log q(\tilde{x})\right),$$

where line 2 to line 3 follows Lemma 1 and line 4 to line 5 follows Lemma 2. $\square$

It is noteworthy that line 3 in the proof of Lemma 3 also showcases the connection between the covariance of $q(\tilde{x}|x)$ and the score of $q(\tilde{x})$:

$$\mathrm{Cov}_{q(x|\tilde{x})}[x] = \frac{\beta^2}{\alpha}\left(\frac{1}{\beta}\mathbb{E}_{q(x|\tilde{x})}[\epsilon\epsilon^T] - \nabla_{\tilde{x}} \log q(\tilde{x})\nabla_{\tilde{x}} \log q(\tilde{x})^T\right), \tag{32}$$

where $\epsilon = (\tilde{x} - \sqrt{\alpha}x)/\sqrt{\beta}$. Now we can apply Equations (31) and (32) to establish the connection between OCM-DDPM and SN-DDPM, as demonstrated in the following theorem.

**Theorem 3** (Connection between OCM-DDPM and SN-DDPM). *Suppose $q(x_{0:T})$ is defined as Equation (1), and the pre-trained score function is well-learned, i.e., $\nabla_{x_t} \log p_\theta(x_t) = \nabla_{x_t} \log q(x_t), \forall x_t$. Let $h_\phi, g_\psi$ be parameterized neural networks. OCM-DDPM learns $h_\phi$ by minimizing the objective $\mathcal{L}_{OCM}(\phi)$ as in Equation (13), and SN-DDPM learns $g_\psi$ by minimizing the objective $\mathcal{L}_{SN}(\psi)$ as in Equation (22). Then in optimal training with $\phi^* = \arg\min \mathcal{L}_{OCM}(\phi)$ and $\psi^* = \arg\min \mathcal{L}_{SN}(\psi)$, we have the optimal diagonal covariance of OCM-DDPM $\Sigma_{t-1}(x_t; \phi^*) = \frac{(1-\alpha_t)^2 h_{\phi^*}(x_t) + (1-\alpha_t)I}{\alpha_t}$ and that of SN-DDPM $\Sigma_{t-1}(x_t; \psi^*) = \frac{1-\bar{\alpha}_{t-1}}{1-\bar{\alpha}_t}\beta_t + \frac{\beta_t^2}{\alpha_t}\left(\frac{g_{\psi^*}(x_t)}{1-\bar{\alpha}_t} - \nabla_{x_t} \log p_\theta(x_t)^2\right)$ are identical.*

*Proof.* In optimal training, we know that $h_{\phi^*}(x_t) = \mathrm{diag}(\nabla_{x_t}^2 \log q(x_t))$ and $g_{\psi^*}(x_t) = \mathbb{E}_{q(x_0|x_t)}[\epsilon_t^2]$. Bao et al. (2022b) show that the covariance of the denoising density $q(x_{t-1}|x_t)$ has a closed form (see Lemma 13 in Bao et al. (2022b))

$$\mathrm{Cov}_{q(x_{t-1}|x_t)}[x_{t-1}] = \lambda_t^2 I + \gamma_t^2 \mathrm{Cov}_{q(x_0|x_t)}[x_0], \tag{33}$$

where $\lambda_t^2 = \frac{1-\bar{\alpha}_{t-1}}{1-\bar{\alpha}_t}\beta_t$ and $\gamma_t = \sqrt{\bar{\alpha}_{t-1}} - \sqrt{1-\bar{\alpha}_{t-1}-\lambda_t^2}\sqrt{\frac{\bar{\alpha}_t}{1-\bar{\alpha}_t}} = \sqrt{\bar{\alpha}_{t-1}}\frac{\beta_t}{1-\bar{\alpha}_t}$. Since $q(x_t|x_0) = \mathcal{N}(\sqrt{\bar{\alpha}_t}x_0, (1-\bar{\alpha}_t)I)$, applying Lemma 3 gives

$$\text{diag}(\text{Cov}_{q(x_0|x_t)}[x_0]) = \frac{(1-\bar{\alpha}_t)}{\bar{\alpha}_t}(I + (1-\bar{\alpha}_t)h_{\phi^*}(x_t)).$$

Substituting it into Equation (33), we have

$$\text{diag}(\text{Cov}_{q(x_{t-1}|x_t)}[x_{t-1}]) = \frac{(1-\alpha_t)^2 h_{\phi^*}(x_t) + (1-\alpha_t)I}{\alpha_t} = \Sigma_{t-1}(x_t; \phi^*)$$

as desired. Alternatively, applying Equation (32), we have

$$\text{diag}(\text{Cov}_{q(x_0|x_t)}[x_0]) = \frac{(1-\bar{\alpha}_t)^2}{\bar{\alpha}_t}\left(\frac{g_{\psi^*}(x_t)}{1-\bar{\alpha}_t} - \nabla_{x_t}\log p_\theta(x_t)^2\right).$$

Substituting it into Equation (33) gives

$$\text{diag}(\text{Cov}_{q(x_{t-1}|x_t)}[x_{t-1}]) = \frac{1-\bar{\alpha}_{t-1}}{1-\bar{\alpha}_t}\beta_t + \frac{\beta_t^2}{\alpha_t}\left(\frac{g_{\psi^*}(x_t)}{1-\bar{\alpha}_t} - \nabla_{x_t}\log p_\theta(x_t)^2\right) = \Sigma_{t-1}(x_t; \psi^*).$$

Therefore, $\text{diag}(\text{Cov}_{q(x_{t-1}|x_t)}[x_{t-1}]) = \Sigma_{t-1}(x_t; \phi^*) = \Sigma_{t-1}(x_t; \psi^*)$ as desired. □

**Remark.** *Theorem 3 establishes the connection between OCM-DDPM and SN-DDPM using the second-order Tweedie's formula. This connection allows OCM-DDPM to utilize the same covariance clipping trick as in SN-DDPM (see Appendix B.2 for details). Additionally, as highlighted in Bao et al. (2022a), SN-DDPM suffers from error amplification in the quadratic term, a limitation not present in OCM-DDPM. As shown in Figure 4 and Table 3, OCM-DDPM demonstrates superior covariance estimation accuracy and likelihood performance compared to SN-DDPM, underscoring the advantages of the proposed optimal covariance matching objective.*

## B  DETAILS OF EXPERIMENTS

### B.1  DETAILS OF MODEL ARCHITECTURES

In this section, we describe our model architectures in detail. Following the approach in Bao et al. (2022a), our model comprises a pretrained score neural network with fixed parameters, along with a trainable diagonal Hessian prediction network built on top of it.

**Details of Pretrained Score Prediction Networks.** Table 6 lists the pretrained neural networks utilized in our experiments. These models parameterize the noise prediction $\epsilon_\theta(x)$, allowing us to derive the score prediction as $s_\theta(x_t) = \nabla_x \log p_\theta(x_t) = -\frac{\epsilon_\theta(x_t)}{\sqrt{1-\bar{\alpha}_t}}$, following the forward process defined in Equation (1). It is important to note that the pretrained networks for ImageNet 64x64 and 256x256 include both noise prediction networks and covariance networks. In our model architectures, we utilize only the noise prediction networks.

Table 6: Source of pretrained score prediction networks used in our experiments.

|  | PROVIDED BY |
| --- | --- |
| CIFAR10 (LS) | BAO ET AL. (2022B) |
| CIFAR10 (CS) | BAO ET AL. (2022B) |
| CELEBA 64X64 | SONG ET AL. (2020) |
| IMAGENET 64X64 | NICHOL & DHARIWAL (2021) |
| LSUN BEDROOM | HO ET AL. (2020) |
| IMAGENET 256X256 | PEEBLES & XIE (2023) |

**Details of Diagonal Hessian Prediction Networks.** For fair comparisons, we follow the parameterization as per Bao et al. (2022a) for all models excluding the one on ImageNet 256x256, which was not explored in their paper. The architecture details of $NN_1$ and $NN_2$ are provided in Table 7, where Conv denotes the convolutional layer, Res denotes the residual block

Table 7: Architecture details of our models.

|  | $NN_1$ | $NN_2$ |
| --- | --- | --- |
| CIFAR10 (LS) | Conv | Conv |
| CIFAR10 (CS) | Conv | Conv |
| CELEBA 64X64 | Conv | Conv |
| IMAGENET 64X64 | Conv | Res+Conv |
| LSUN BEDROOM | Conv | Res+Conv |
| IMAGENET 256X256 | AdaLN+Linear | AdaLN+Linear |

Table 8: The number of parameters and the averaged time (ms) to run a model function evaluation. All are evaluated with a batch size of 64 on one A100-80GB GPU.

| | SCORE PREDICTION NETWORK | SCORE & SN PREDICTION NETWORKS | SCORE & DIAGONAL HESSIAN PREDICTION NETWORKS |
|---|---|---|---|
| CIFAR10 (LS) | 52.54 M / 44.37 MS | 52.55 M / 44.61 MS (+0.5%) | 52.55 M / 44.51 MS (+0.3%) |
| CIFAR10 (CS) | 52.54 M / 45.13 MS | 52.55 M / 45.24 MS (+0.2%) | 52.55 M / 45.23 MS (+0.2%) |
| CELEBA 64x64 | 78.70 M / 67.88 MS | 78.71 M / 69.15 MS (+1.9%) | 78.71 M / 68.89 MS (+1.5%) |
| IMAGENET 64x64 | 121.06 M / 106.58 MS | 121.49 M / 112.53 MS (+5.6%) | 121.49 M / 112.23 MS (+5.3%) |
| LSUN BEDROOM | 113.67 M / 692.58 MS | 114.04 M / 771.55 MS (+11.4%) | 114.04 M / 770.73 MS (+11.2%) |
| IMAGENET 256x256 | 675.13 M / 22.84 MS | MISSING MODEL | 677.80 M / 23.01 MS (+0.7%) |

(He et al., 2016), AdaLN denotes the adaptive layer norm block (Peebles & Xie, 2023), and Linear denotes the linear layer.

**Cost of Memory and Inference Time.** In Table 8, we present the number of parameters and inference time for various models. It is evident that the additional memory cost of the diagonal Hessian prediction network is minimal compared to the original diffusion models, which only include the score prediction. Regarding the extra inference cost, it is negligible for CIFAR10, CelebA 64x64, and ImageNet 256x256. The additional time is at most $5.3\%$ on ImageNet 64x64, and $11.2\%$ on LSUN Bedroom, but this is offset by the benefit of requiring fewer sampling steps to achieve an FID around 6, as shown in Table 4. Notably, although the model size on ImageNet 256x256 is the largest, it has the shortest inference time because DiT learns the diffusion model in the latent space.

**Details of Table 4.** In Table 4, the FID results of baselines are taken from Bao et al. (2022a). Notably, the time cost for a single neural function evaluation is identical across DDPM, DDIM, and Analytic-DPM, all of which use a fixed isotropic covariance in the backward Markov process. The ratio of the time cost to the baselines is based on Table 8. For the FID results of our model, OCM-DPM, we report performance using the DDPM forward process for LSUN Bedroom and the DDIM forward process for the other datasets.

### B.2 DETAILS OF TRAINING, INFERENCE, AND EVALUATION

Our training and inference recipes largely follow those outlined in Bao et al. (2022a); Peebles & Xie (2023). Below, we provide a detailed description.

**Training Details.** We use the AdamW optimizer (Loshchilov, 2017) with a learning rate of 0.0001 and train for 500K iterations across all datasets. The batch sizes are set to 64 for LSUN Bedroom, 128 for CIFAR10, CelebA 64x64, and ImageNet 64x64, and 256 for ImageNet 256x256. During training, checkpoints are saved every 10K iterations, and we select the checkpoint with the best FID on 2048 samples generated with full sampling steps[2]. We train our models using one A100-80G GPU for CIFAR10, CelebA 64x64, and ImageNet 64x64; four A100-80G GPUs for LSUN Bedroom; and eight A100-80G GPUs for ImageNet 256x256.

**Sampling Details.** As per Bao et al. (2022b), covariance clipping is crucial to the performance in diffusion models with unfixed variance in the backward Markovian. Leveraging the connection between OCM-DPM and SN-DPM established in Theorem 3, we can apply the same clipping strategies outlined in Bao et al. (2022a;b). Specifically, we only display the mean of $p(x_0|x_1)$ at the last sampling and clip the covariance $\Sigma_1(x_2)$ of $p(x_1|x_2)$ such that $\|\Sigma_1(x_2)\|_\infty \mathbb{E}|\epsilon| \leq \frac{2}{255}y$, where $\|\cdot\|_\infty$ denotes the infinity norm and $\epsilon$ is the standard Gaussian noise. Following Bao et al. (2022b), we use $y = 2$ on CIFAR10 (LS) and CelebA 64x64 under the DDPM forward process, and use $y = 1$ for other cases. For all skip-step sampling methods, we employ the even trajectory (Nichol & Dhariwal, 2021) for selecting the subset of sampling steps.

**Evaluation Details.** The performance is evaluated on the exponential moving average (EMA) model with a rate of 0.9999. For computing the negative log-likelihood, we follow Ho et al. (2020); Bao et al. (2022a) by discretizing the last sampling step $p(x_0|x_1)$ to obtain the likelihood of discrete image data and report the upper bound of the negative log-likelihood on the entire test set. The FID score is computed on 50K generated samples. Following Nichol & Dhariwal (2021); Bao et al.

---

[2]We use 2048 samples instead of 1000 to ensure that the covariance of the Inception features has full rank.

Table 9: The corresponding mean and standard deviation of NLL and FID reported in Tables 3 and 5. Note that the standard deviation is reported under the scale of percentage (%).

| | CIFAR10 (LS) | | | | | | CIFAR10 (CS) | | | | | |
|---|---|---|---|---|---|---|---|---|---|---|---|---|
| # TIMESTEPS $K$ | 10 | 25 | 50 | 100 | 200 | 1000 | 10 | 25 | 50 | 100 | 200 | 1000 |
| MEAN-DDPM (NLL) | 5.33 | 4.63 | 4.35 | 3.97 | 3.78 | 3.57 | 4.99 | 4.34 | 3.99 | 3.76 | 3.59 | 3.41 |
| STD-DDPM (NLL) % | 0.04 | 0.03 | 0.03 | 0.04 | 0.04 | 0.01 | 0.02 | 0.01 | 0.03 | 0.03 | 0.02 | 0.01 |
| MEAN-DDPM (FID) | 25.07 | 9.30 | 5.90 | 4.37 | 3.54 | 4.00 | 14.46 | 5.49 | 4.09 | 3.83 | 3.81 | 4.22 |
| STD-DDPM (FID) % | 9.05 | 7.54 | 3.48 | 2.77 | 4.25 | 2.30 | 10.40 | 7.74 | 2.66 | 1.32 | 5.26 | 3.08 |
| MEAN-DDIM (FID) | 10.61 | 4.31 | 3.48 | 3.26 | 3.25 | 3.75 | 16.85 | 6.70 | 4.73 | 4.33 | 4.56 | 4.59 |
| STD-DDIM (FID) % | 5.17 | 3.55 | 3.19 | 1.87 | 3.97 | 0.99 | 13.22 | 4.23 | 2.28 | 2.94 | 1.61 | 5.30 |

| | CELEBA 64x64 | | | | | | IMAGENET 64x64 | | | | | |
|---|---|---|---|---|---|---|---|---|---|---|---|---|
| # TIMESTEPS $K$ | 10 | 25 | 50 | 100 | 200 | 1000 | 25 | 50 | 100 | 200 | 400 | 4000 |
| MEAN-DDPM (NLL) | 4.69 | 3.86 | 3.43 | 3.13 | 2.90 | 2.66 | 4.45 | 4.15 | 3.93 | 3.79 | 3.70 | 3.59 |
| STD-DDPM (NLL) % | 0.01 | 0.01 | 0.01 | 0.00 | 0.01 | 0.00 | 0.01 | 0.00 | 0.01 | 0.00 | 0.01 | 0.01 |
| MEAN-DDPM (FID) | 21.58 | 12.65 | 9.24 | 6.96 | 6.00 | 5.02 | 28.01 | 20.85 | 18.01 | 16.76 | 16.35 | 16.33 |
| STD-DDPM (FID) % | 2.24 | 7.65 | 2.29 | 2.32 | 5.50 | 1.37 | 0.77 | 4.66 | 6.82 | 3.28 | 3.04 | 3.21 |
| MEAN-DDIM (FID) | 10.36 | 5.69 | 4.37 | 3.50 | 3.11 | 3.03 | 28.34 | 19.68 | 17.84 | 17.41 | 17.36 | 18.91 |
| STD-DDIM (FID) % | 7.40 | 4.04 | 5.12 | 4.88 | 6.46 | 1.55 | 6.59 | 3.77 | 9.15 | 4.08 | 7.88 | 5.26 |

(a) Cov Error with True Score   (b) Cov Error with Learned Score   (c) DDPM MMD v.s. Steps   (d) DDIM MMD v.s. Steps

Figure 4: Comparisons of different covariance estimation methods based on estimation error and sample generation quality. Figures (a) and (b) show the mean square error of the estimated diagonal covariance under the assumptions: (a) access to the true score, and (b) learned score of the data distribution at various noise levels. Figures (c) and (d) present the MMD evaluation against the total sampling steps in the DDPM (c) and DDIM (d) settings. We can find the proposed OCM method can achieve the lowest estimation error and consistently outperform other baseline methods when fewer generation steps are applied.

(2022a); Peebles & Xie (2023), the reference distribution statistics for FID are calculated using the full training set for CIFAR10 and ImageNet, and 50K training samples for CelebA and LSUN Bedroom. Performance results are reported using the same random seed[3] as in Bao et al. (2022a). Additionally, the variance across different random seeds is provided in Appendix C.4.

## C   ADDITIONAL EXPERIMENTAL RESULTS

In this section, we present additional experimental results for comparison. First, we present an additional toy experiment in Appendix C.1 and likelihood comparison between I-DDPM, NPR-DDPM, and OCM-DDPM in Appendix C.2 and present results using different numbers of Rademacher samples in Appendix C.3. We also analyze the performance variance in Appendix C.4. Additionally, we include further experimental results on latent diffusion models in Appendix C.5 and conduct qualitative studies by showcasing samples generated by our models in Appendix C.6.

### C.1   ADDITIONAL TOY DEMONSTRATION

To further verify the effectiveness of our method, we include an additional toy example. In this case, we consider another two-dimensional mixture of nine Gaussians (MoG) with means located at

---

[3]We use the official code from https://github.com/baoffff/Extended-Analytic-DPM.

$\{-3, 0, 3\} \otimes \{-3, 0, 3\}$ and a standard deviation of $\sigma = 0.1$. To assess different approaches, we first learn the covariance using the true scores. Specifically, we train the covariance networks for these methods over 50,000 iterations with a learning rate of 0.001 using an Adam optimizer (Kingma & Ba, 2014). Figure 4a shows the $L_2$ error of the estimated diagonal covariance $\Sigma_{t-1}(x_t)$, and our method, OCM-DDPM, consistently achieves the lowest error compared to the other methods. In practice, the true score is not accessible. Therefore, we also conduct a comparison using a score function learned with DSM. We then apply different covariance estimation methods under the same settings. In Figure 4b, we plot the same $L_2$ error for the learned score setting and find that our method achieves the lowest error for all $t$ values.

We further use the learned score and the covariance estimated by different methods to conduct DDPM and DDIM sampling for this MoG problem. Figures 4c and 4d presents the MMD comparison across various methods. It shows that our method outperforms the baselines when the total time step is small, demonstrating the importance of accurate covariance estimation in the context of diffusion acceleration. We also include two methods utilizing the true diagonal and full covariance as benchmarks, representing the best achievable performance. Notably, these two methods exhibit similar performance because the MoG used in this setting has symmetric components, in which the covariance is dominated by the diagonal entries.

## C.2 ADDITIONAL LIKELIHOOD COMPARISON

Here, we provide additional likelihood comparisons against I-DDPM (Nichol & Dhariwal, 2021) on the ImageNet 64x64 dataset. As discussed in Section 4, I-DDPM parameterizes the diagonal covariance by interpolating between $\beta$ and and learns the covariance by maximizing the variational lower bound. For ease of comparison, we also include the performance of NPR-DDPM and SN-DDOM, which employ an alternative MSE loss as detailed in

Table 10: Upper bound on the negative log-likelihood (bits/dim) on the ImageNet 64x64 dataset.

| # TIMESTEPS $K$ | 25 | 50 | 100 | 200 | 400 | 4000 |
|---|---|---|---|---|---|---|
| I-DDPM | 18.91 | 8.46 | 5.27 | 4.24 | 3.86 | **3.57** |
| NPR-DDPM | 4.66 | 4.22 | 3.96 | 3.80 | 3.71 | 3.60 |
| SN-DDPM | 4.56 | 4.18 | 3.95 | 3.80 | 3.71 | 3.63 |
| OCM-DDPM | **4.45** | **4.15** | **3.93** | **3.79** | **3.70** | 3.59 |

Equation (22) to learn the covariance. Table 10 presents the results, showing that OCM-DDPM achieves the best likelihood with fewer sampling steps while maintaining performance comparable to I-DDPM when using full sampling steps. This highlights the superiority of the proposed optimal covariance matching objective in learning a diffusion model for data density estimation.

## C.3 IMPACT OF THE NUMBERS OF RADEMACHER SAMPLES

In this section, we include an ablation study examining the effect of varying the number of Rademacher samples $M$. Specifically, we conduct experiments on CIFAR10 (CS) using the proposed OCM-DDPM method and evaluate performance based on FID and NLL. The empirical results, presented in Table 11, indicate that a larger $M$ (e.g., $M = 3$) can give a small improvement in FID. This improvement is likely due to the reduced gradient estimation variance during training with a larger $M$. However,

Table 11: Results of OCM-DDPM on CIFAR10 (CS) with varying numbers of Rademacher Samples.

| | FID ↓ | | | | NLL (%) ↑ | | | |
|---|---|---|---|---|---|---|---|---|
| $K =$ | 10 | 25 | 50 | 100 | 10 | 25 | 50 | 100 |
| $M = 1$ | 14.32 | 5.54 | 4.10 | 3.84 | 4.99 | 4.34 | 3.99 | 3.76 |
| $M = 3$ | 14.18 | 5.51 | 4.11 | 3.82 | 4.99 | 4.34 | 3.99 | 3.76 |
| $M = 5$ | 14.17 | 5.51 | 4.11 | 3.82 | 4.99 | 4.34 | 3.99 | 3.76 |
| $M = 10$ | 14.16 | 5.51 | 4.11 | 3.82 | 4.99 | 4.34 | 3.99 | 3.76 |

in most cases, different values of $M$ yield consistent performance. This is practically desirable as, in practice, setting $M = 1$ allows for efficient training while maintaining strong performance

## C.4 MEAN AND VARIANCE OF PERFORMANCE

In Tables 3 and 5, we evaluate our models using the same random seed as Bao et al. (2022a). To minimize the impact of randomness, we report the mean and standard deviation in Table 9 by repeating the evaluation three times with different seeds.

Table 12: Results with varying CFG coefficients using 10 sampling steps on ImageNet 256x256.

| | FID ↓ | | | | | RECALL (%) ↑ | | | | |
| CFG = | 1.5 | 1.75 | 2.0 | 3.0 | 4.0 | 1.5 | 1.75 | 2.0 | 3.0 | 4.0 |
|---|---|---|---|---|---|---|---|---|---|---|
| DDPM, $\tilde{\beta}$ | 30.41 | 19.26 | 13.53 | 11.52 | 14.76 | 39.74 | 35.29 | 32.08 | 24.93 | 19.91 |
| DDPM, $\beta$ | 210.28 | 182.78 | 158.16 | 89.86 | 58.16 | 16.98 | 22.28 | 25.13 | 23.07 | 19.48 |
| I-DDPM | 44.96 | 20.71 | 13.01 | 9.76 | 13.75 | 50.32 | 43.75 | 41.08 | 31.77 | 25.49 |
| OCM-DDPM | 30.55 | 17.57 | 11.12 | 9.52 | 13.74 | 49.39 | 45.87 | 42.33 | 32.13 | 24.96 |
| DDIM | 9.41 | 6.54 | 6.12 | 10.49 | 14.18 | 49.15 | 45.82 | 41.15 | 29.27 | 23.16 |
| OCM-DDIM | 11.30 | 6.67 | 5.33 | 9.20 | 13.49 | 54.50 | 50.32 | 46.58 | 34.53 | 25.67 |

Table 13: Results with CFG=1.5 across different sampling steps on ImageNet 256x256.

| | FID ↓ | | | | | | RECALL (%) ↑ | | | | | |
| # TIMESTEPS $K$ | 10 | 25 | 50 | 100 | 200 | 250 | 10 | 25 | 50 | 100 | 200 | 250 |
|---|---|---|---|---|---|---|---|---|---|---|---|---|
| DDPM, $\tilde{\beta}$ | 30.41 | 4.79 | 3.55 | 3.09 | 3.05 | 2.97 | 39.74 | 49.02 | 51.79 | 53.54 | 54.21 | 54.34 |
| DDPM, $\beta$ | 210.28 | 42.09 | 9.43 | 3.63 | 2.91 | 2.75 | 16.98 | 46.76 | 51.67 | 55.01 | 55.50 | 55.21 |
| I-DDPM | 44.96 | 9.01 | 3.70 | 2.48 | 2.25 | 2.74 | 50.32 | 54.78 | 56.45 | 57.88 | 58.76 | 54.69 |
| OCM-DDPM | 30.55 | 4.96 | 3.21 | 2.71 | 2.50 | 2.75 | 49.39 | 53.75 | 54.68 | 55.85 | 54.97 | 54.75 |
| DDIM | 9.41 | 2.70 | 2.33 | 2.25 | 2.23 | 2.20 | 49.15 | 56.41 | 57.00 | 57.83 | 57.81 | 57.99 |
| OCM-DDIM | 11.30 | 3.26 | 2.56 | 2.28 | 2.23 | 2.18 | 54.50 | 58.33 | 58.09 | 58.80 | 58.16 | 58.56 |

## C.5 MORE RESULTS ON LATENT DIFFUSION MODELS

We report the performance using 10 sampling steps with varying CFG coefficients in Table 12. The results indicate that our methods perform best at CFG = 2.0. While DDPM-$\tilde{\beta}$ and DDIM show strong FID scores, their Recall is lower due to fixed variance, suggesting less diversity in the generated samples. In Table 13, we further compare the performance across different sampling steps at CFG = 1.5. The results again demonstrate that our methods, which estimate the optimal covariance from the data, produce more diverse samples while maintaining comparable image quality.

## C.6 GENERATED SAMPLES

In this section, we conduct qualitative studies by showcasing the generated samples from our models using different sampling steps $K$. The results are summarized as follows:

- In Figure 6, we visualize the generated samples using varying numbers of Monte Carlo samples with the Rademacher estimator (refer to Table 1).

- In Figure 7, we visualize the training data and the generated samples using the minimum number of sampling steps required to achieve an FID of approximately 6 (refer to Table 4).

- In Figures 8 to 11, we visualize the generated samples of our models using different number of sampling steps on CIFAR10 (LS), CIFAR10 (CS), CelebA 64x64, and ImageNet 64x64, respectively (refer to Table 5).

- In Figure 12, we visualize the generated samples on ImageNet 256x256, using 10 sampling steps with different CFG coefficients. (refer to Figure 3 and Table 12).

- In Figure 13, we visualize the generated samples on ImageNet 256x256, using varying number of sampling steps with CFG set to 1.5 (refer to Table 13).

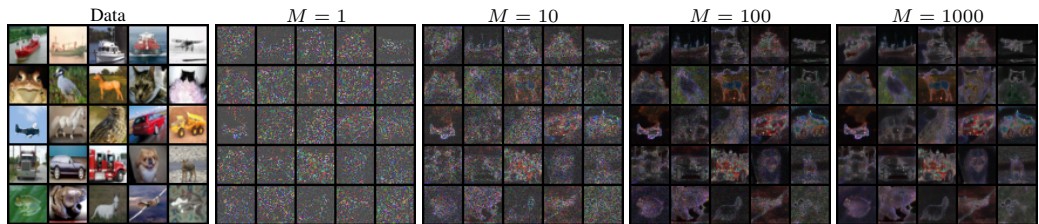

Figure 5: Diagonal covariance estimation visualisation with different Rademacher sample numbers.

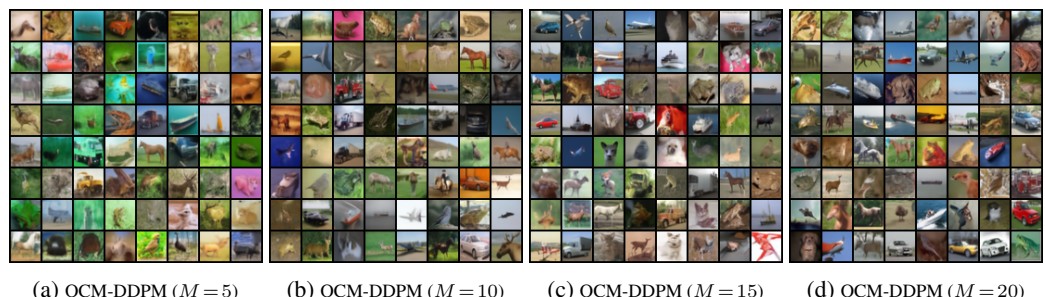

(a) OCM-DDPM ($M = 5$)    (b) OCM-DDPM ($M = 10$)    (c) OCM-DDPM ($M = 15$)    (d) OCM-DDPM ($M = 20$)

Figure 6: Generated samples with 10 sampling steps using ifferent Rademacher sample numbers on CIFAR10 (CS) (ref: Table 1).

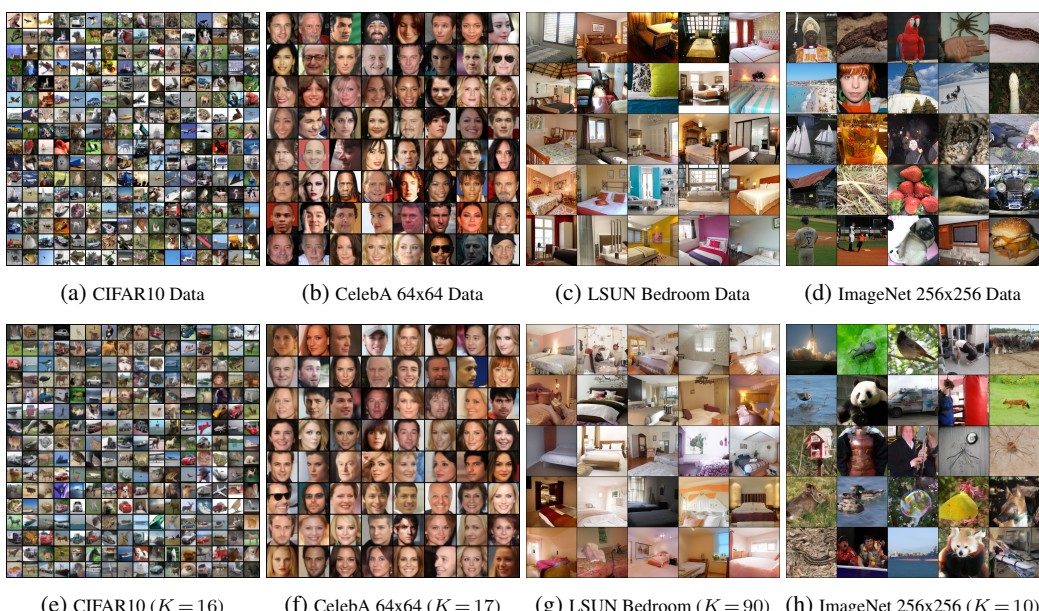

(a) CIFAR10 Data    (b) CelebA 64x64 Data    (c) LSUN Bedroom Data    (d) ImageNet 256x256 Data

(e) CIFAR10 ($K = 16$)    (f) CelebA 64x64 ($K = 17$)    (g) LSUN Bedroom ($K = 90$)    (h) ImageNet 256x256 ($K = 10$)

Figure 7: The training data and generated samples of OCM-DPM with minimum steps to achieve an FID around 6. (ref: Table 4)

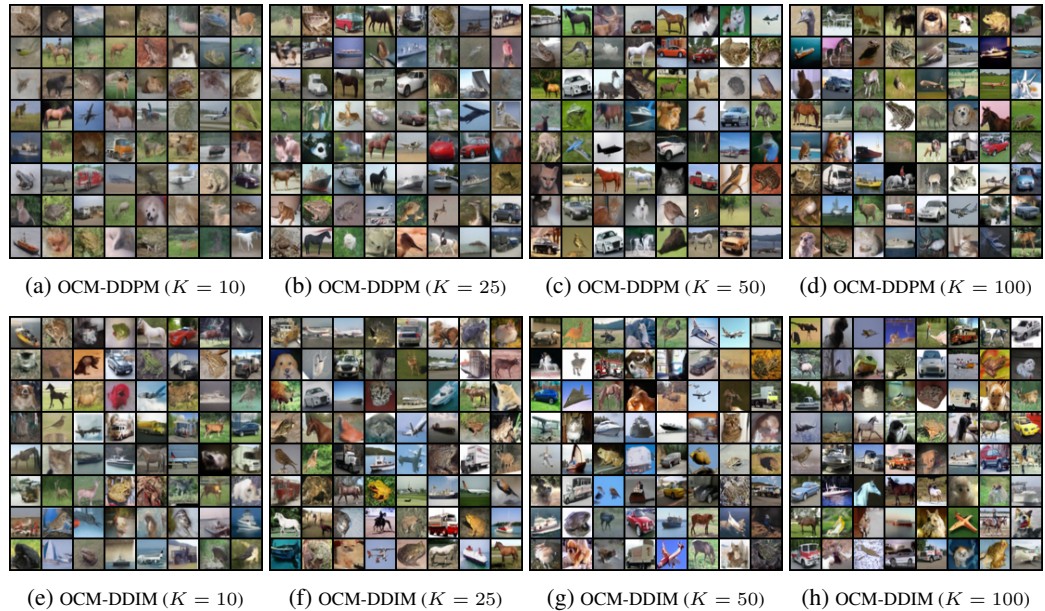

(a) OCM-DDPM ($K = 10$)     (b) OCM-DDPM ($K = 25$)     (c) OCM-DDPM ($K = 50$)     (d) OCM-DDPM ($K = 100$)

(e) OCM-DDIM ($K = 10$)     (f) OCM-DDIM ($K = 25$)     (g) OCM-DDIM ($K = 50$)     (h) OCM-DDIM ($K = 100$)

Figure 8: Generated samples with different sampling steps on CIFAR10 (LS) (ref: Table 5).

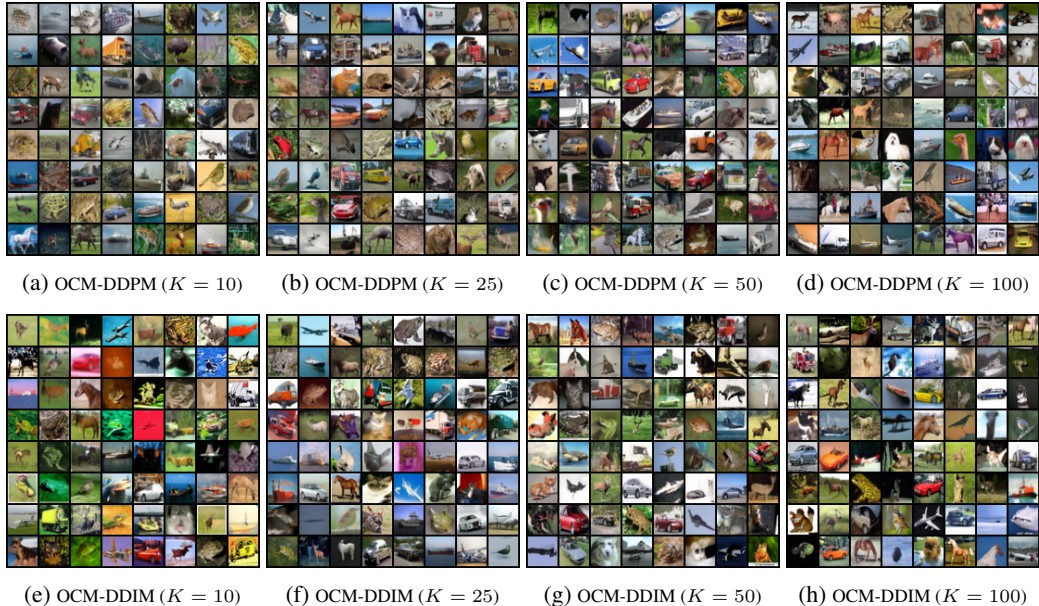

(a) OCM-DDPM ($K = 10$)     (b) OCM-DDPM ($K = 25$)     (c) OCM-DDPM ($K = 50$)     (d) OCM-DDPM ($K = 100$)

(e) OCM-DDIM ($K = 10$)     (f) OCM-DDIM ($K = 25$)     (g) OCM-DDIM ($K = 50$)     (h) OCM-DDIM ($K = 100$)

Figure 9: Generated samples with different sampling steps on CIFAR10 (CS) (ref Table 5).

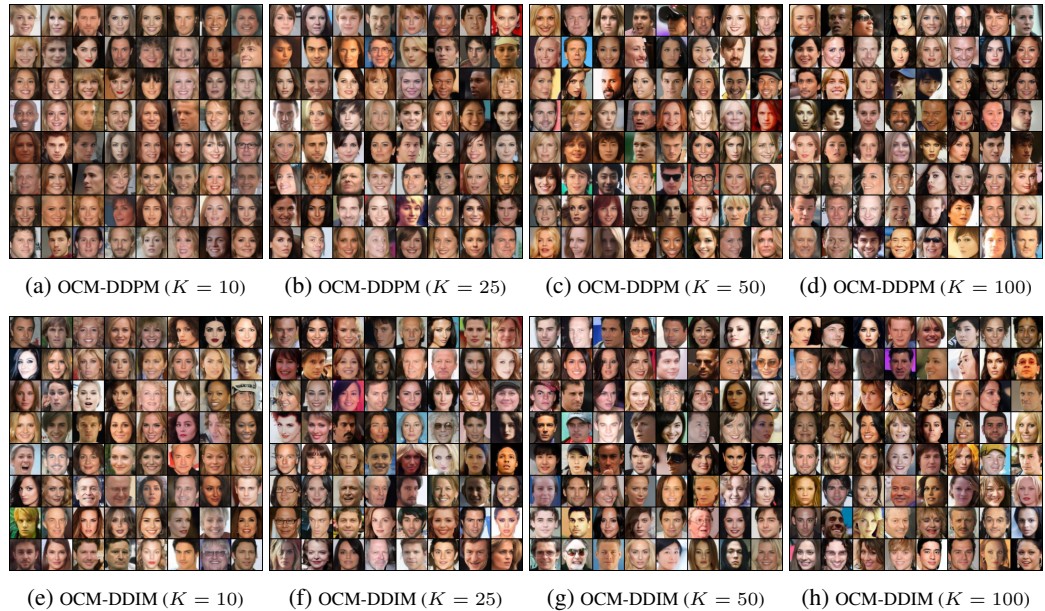

(a) OCM-DDPM ($K = 10$)    (b) OCM-DDPM ($K = 25$)    (c) OCM-DDPM ($K = 50$)    (d) OCM-DDPM ($K = 100$)

(e) OCM-DDIM ($K = 10$)    (f) OCM-DDIM ($K = 25$)    (g) OCM-DDIM ($K = 50$)    (h) OCM-DDIM ($K = 100$)

Figure 10: Generated samples with different sampling steps on CelebA 64x64 (ref: Table 5).

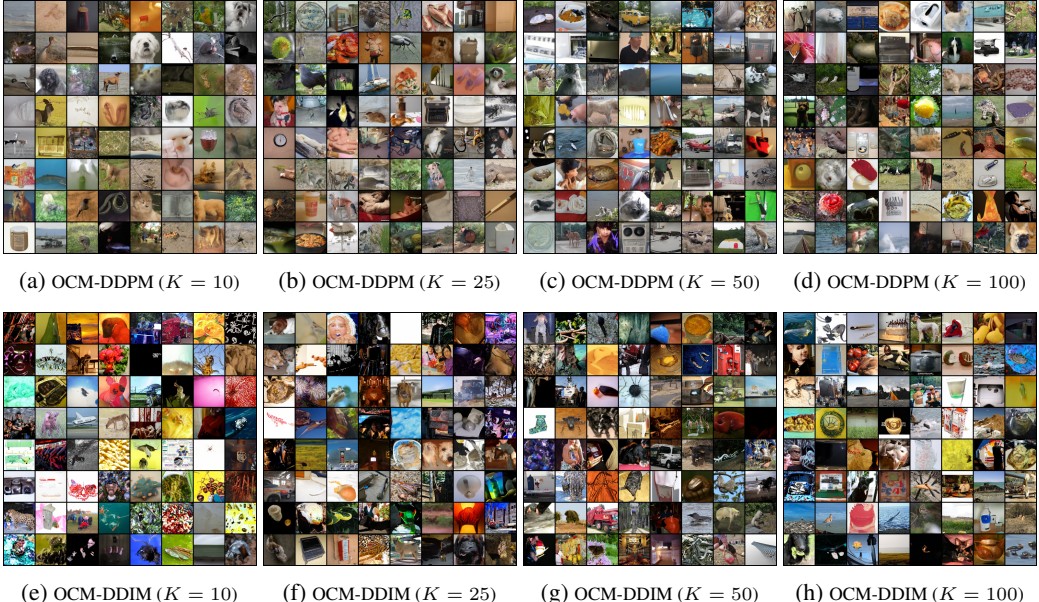

(a) OCM-DDPM ($K = 10$)    (b) OCM-DDPM ($K = 25$)    (c) OCM-DDPM ($K = 50$)    (d) OCM-DDPM ($K = 100$)

(e) OCM-DDIM ($K = 10$)    (f) OCM-DDIM ($K = 25$)    (g) OCM-DDIM ($K = 50$)    (h) OCM-DDIM ($K = 100$)

Figure 11: Generated samples with different sampling steps on ImageNet 64x64 (ref: Table 5).

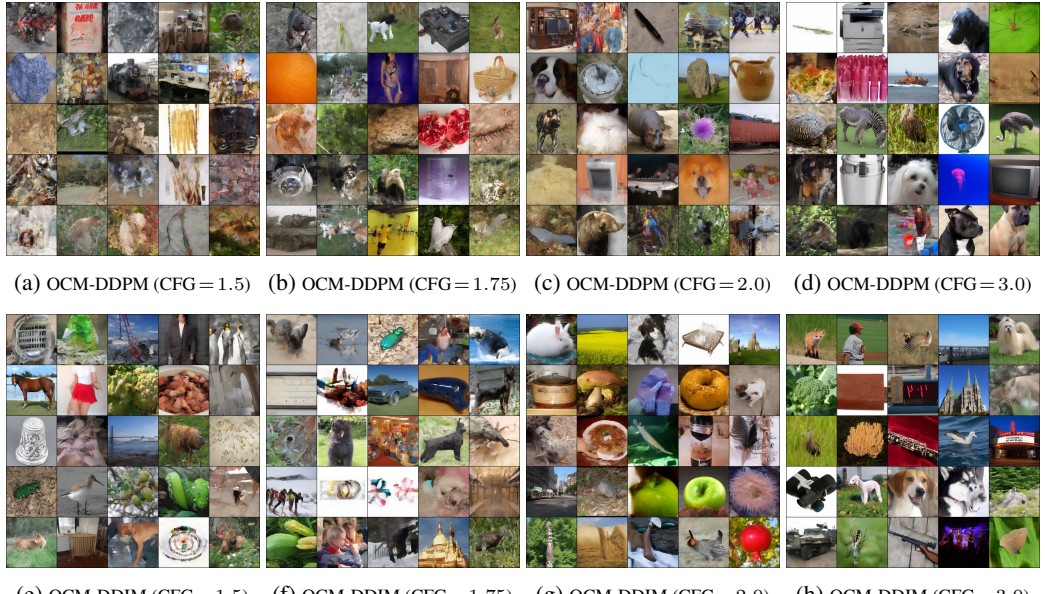

(a) OCM-DDPM (CFG = 1.5)   (b) OCM-DDPM (CFG = 1.75)   (c) OCM-DDPM (CFG = 2.0)   (d) OCM-DDPM (CFG = 3.0)

(e) OCM-DDIM (CFG = 1.5)   (f) OCM-DDIM (CFG = 1.75)   (g) OCM-DDIM (CFG = 2.0)   (h) OCM-DDIM (CFG = 3.0)

Figure 12: Generated samples with 10 sampling steps using different CFG coefficients on ImageNet 256x256 (ref: Table 12)

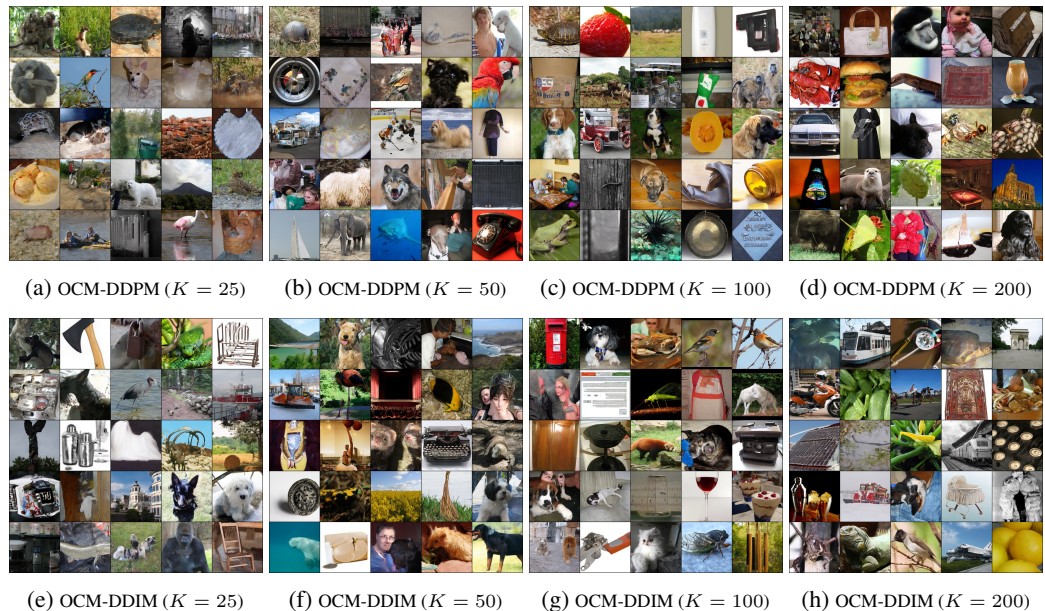

(a) OCM-DDPM ($K = 25$)   (b) OCM-DDPM ($K = 50$)   (c) OCM-DDPM ($K = 100$)   (d) OCM-DDPM ($K = 200$)

(e) OCM-DDIM ($K = 25$)   (f) OCM-DDIM ($K = 50$)   (g) OCM-DDIM ($K = 100$)   (h) OCM-DDIM ($K = 200$)

Figure 13: Generated samples with CFG=1.5 using different sampling steps on ImageNet 256x256 (ref: Table 13)

