# OpenReview forum: "Improving Probabilistic Diffusion Models With Optimal Diagonal Covariance Matching"
_ICLR.cc/2025/Conference — ICLR 2025 Oral_

### Official Review · Reviewer_BQbT · 2024-10-21

**Soundness:** 3
**Presentation:** 4
**Contribution:** 2
**Rating:** 8
**Confidence:** 3

**Summary:**

The authors propose an $x_t$-dependent diagonal covariance estimation method that directly learns from score function. This leads to improved efficiency and more stable/robust denoising, which in turn improves the image generation quality.

**Strengths:**

1. Very clear and comprehensive presentation, easy to read and follow.
2. The proposed method is a general method that applies to various settings.
3. The description of experimental details is generally comprehensive.

**Weaknesses:**

1. The idea seems both incremental and way-too simple, which is not a fundamental change in the covariance estimation that would lead to a revolutionized/big change in the performance and/or efficiency. It will have a limited impact on the community.

2. The experimental results seem not extraordinary. It does not fundamentally improve the performance (generation quality), which is understandable since covariance is more about sampling efficiency and sample diversity. However, even in terms of the sampling diversity and efficiency, such as indicated by Table 4, the proposed method does not significantly beat strong baselines. Moreover, there are no error bars or uncertainty in the experimental results.

**Questions:**

I have no more questions. I like the presentation (as well as the soundness of the idea, of course) and think this alone makes the paper okay to accept. I am just simply not impressed by the results.

---

> ### Author Response · Authors · 2024-11-19
> **Response to Reviewer**
>
> Thanks for your comments and the favourable on our presentation! Here are our responses to your concerns.
>
> > The idea seems both incremental and way-too simple, which is not a fundamental change in the covariance estimation that would lead to a revolutionized/big change in the performance and/or efficiency. It will have a limited impact on the community.
> >
>
> **ANSWER:**
>
> Thank you for your valuable feedback. While our method is simple, it is also highly effective:
>
> 1. Compared to other diffusion models using heuristic or state-independent isotropic variance, our approach consistently outperforms in both FID and likelihood.
> 2. Compared to previous methods for diagonal covariance learning, some achieve better FID but worse likelihood (e.g., SN-DDPM), while others achieve better likelihood but worse FID (e.g., NPR-DDPM). In contrast, our method strikes a superior trade-off between likelihood and FID.
>
> We believe the proposed method has the potential for a broader impact on the community. While FID is a widely used metric for generation quality, many applications of generative models—such as out-of-distribution (OOD) detection, lossly compression, or AI4Science problems—place greater importance on metrics like diversity or likelihood. Our method consistently improves likelihood and diversity without compromising FID, all while enhancing efficiency. We will clarify this point further in the revised version.
>
> Moreover, the main insight of our paper is to show that improving covariance estimation can lead to better performance in both sample quality (FID) and density estimation (likelihood) with fewer NFEs.  We believe this insight will inspire future research. For example, more advanced covariance approximation methods, such as low-rank approximations, as suggested by the reviewer, represent a promising direction for future work.
>
> > The experimental results seem not extraordinary. It does not fundamentally improve the performance (generation quality), which is understandable since covariance is more about sampling efficiency and sample diversity. However, even in terms of the sampling diversity and efficiency, such as indicated by Table 4, the proposed method does not significantly beat strong baselines.
> >
>
> **ANSWER:** Thank you for your comment. While the proposed method does not significantly outperform strong baselines, we emphasize that it achieves a better trade-off between FID and diversity/likelihood with fewer NFEs. We have included an additional visualization to better highlight this benefit on CIFAR10 (CS) (please refer to [this figure](https://imgur.com/a/fid-nll-trade-off-8ctJg8S) for details).
>
> Moreover, as shown in Figure 4, although our method achieves similar FID performance to DDIM, it notably enhances the diversity measured by recall. Additionally, as suggested by reviewers mSjs and tC1T, we expect our work will inspire future research to explore more advanced approximations of the optimal covariance, such as lower-rank or block-diagonal approximations. While these approaches may introduce additional computational costs, we believe they have the potential to achieve better performance. Developing efficient training algorithms for such advanced covariance matching methods is a promising direction for future work.
>
> > Moreover, there are no error bars or uncertainty in the experimental results.
> >
>
> **ANSWER:** Thanks for raising this point, we put the error bar result in Table 9 in the Appendix. We will further highlight this in the main text to improve the rigorousness.

---

> ### Comment · Reviewer_BQbT · 2024-11-19
>
> Thanks for the detailed response. I raised my score. Please include the mentioned important experimental results and visualization in the main paper. Please open-source your code if the paper is accepted.

---

> > ### Author Response · Authors · 2024-11-19
> > **Thanks for your favor of our work**
> >
> > Thank you for taking the time to review our responses and for your appreciation of our work. We are currently refining the paper accordingly, and new results and visualizations will be included in the revision. The code will be released upon the paper’s acceptance.

---

### Official Review · Reviewer_hfvy · 2024-10-28

**Soundness:** 3
**Presentation:** 4
**Contribution:** 3
**Rating:** 8
**Confidence:** 3

**Summary:**

The authors introduce an optimal covariance matching objective based on score functions from pertained diffusion models to learn the covariance of the Gaussian transition kernel.
The proposed method is superior at covariance estimation in terms of  less storage consumption and less number of network evaluations.
Empirical results indicate that the proposed objective enhances performance in both sampling quality (NLL, FID) and efficiency (number of sample steps).

**Strengths:**

- The paper is well-organized, results are clearly presented, and the technique derivations are solid.
- Overall paper organization is very smooth, the methodology part is easy to follow, and experiments are well-designed and coupled with the proposed methodology. e.g. from figure3, table 3 and table 5, I can see that the improved sample quality can be directly attributed from better covariance estimation.
- Unlike the previous works which are of high requirements on storage and network evaluation in covariance estimate (especially the Hessian term), the proposed OCM barely needs 1 network evaluation for each sampling step, so that the method improves number of sampling steps without sacrificing too much sampling speed on covariance estimate.

**Weaknesses:**

- The authors show that the proposed OCM improves covariance estimation for DDPM and DDIM, which results in improved sample quality and better efficiency. There are many other stochastic and deterministic samplers with different sampling schedules, does the improved covariance helps with all these samplers?

**Questions:**

- Is the proposed OCM able to help with other diffusion samplers? e.g. EDM [1] stochastic and deterministic samplers?
- Any insights or intuitions why comparing to baselines, learning the covariance via OCM objective gives better covariance estimation over all noise levels?
- Just out of curiosity, in Fig 3 (a) and (b), why the covariance error goes up for all methods when t is roughly in range [0, 400]?

[1] Karras, Tero, et al. "Elucidating the design space of diffusion-based generative models." Advances in neural information processing systems 35 (2022): 26565-26577.

---

> ### Author Response · Authors · 2024-11-19
> **Response to Reviewer (Part 1)**
>
> Thanks for your favour in our work! Below are our answers to your questions. Please feel free to leave additional comments if you have any further concerns or would like to discuss them further.
>
> > The authors show that the proposed OCM improves covariance estimation for DDPM and DDIM, which results in improved sample quality and better efficiency. There are many other stochastic and deterministic samplers with different sampling schedules, does the improved covariance helps with all these samplers, e.g. EDM [1] stochastic and deterministic samplers?
> >
>
> **ANSWER:** Thank you for your constructive comment. Directly applying the improved covariance to EDM’s stochastic and deterministic samplers is not straightforward because these samplers rely on ODE and SDE solvers. However, our proposed method can be naturally extended to continuous time, allowing it to integrate into the EDM framework when used with DDPM and DDIM samplers.
>
> Specifically, EDM adopts the variance-exploding forward process $p(x_t|x_0) = \mathcal{N}(x_t; x_0, \sigma_t^2 I)$. Since this forward process is Gaussian, it also has an associated optimal covariance in the backward process. Using our proposed optimal covariance matching objective, we can learn the diagonal covariance for this process. During inference, the DDPM or DDIM samplers can leverage the learned covariance for sampling. However, it remains an open question whether this approach can outperform the original EDM sampler or how to integrate the improved covariance into ODE/SDE-based samplers. These are promising directions for future research.
>
> > Any insights or intuitions why comparing to baselines, learning the covariance via OCM objective gives better covariance estimation over all noise levels?
> >
>
> **ANSWER:**
>
> Thank you for your insightful question. We believe it is a very important question, which we should discuss more in the paper to provide more intuitions.  Below, we provide a detailed hypothesis to explain why learning covariance via the OCM objective outperforms baseline methods across all noise levels.
>
> I-DDPM: This method jointly learns the mean and covariance by maximizing the evidence lower bound (ELBO) of the likelihood. We hypothesize that this joint optimization introduces challenges because the covariance estimation depends heavily on the mean. When the mean is suboptimal during training, the ELBO becomes overly loose, resulting in an additional bias in the covariance estimation. In contrast, our method simplifies this process by directly regressing the target covariance, which avoids the complexities of optimizing a loosely bound and leads to a less biased learning procedure.
>
> SN-DDPM:  This method estimates diagonal covariance directly from noisy empirical data. While it improves FID in some cases, it struggles with likelihood estimation. A key issue here is the quadratic term in their model (as discussed in Equation 24 of their paper), which amplifies estimation errors. This effect is particularly pronounced at low noise levels, where score estimation is less accurate. Our method, by directly targeting covariance estimation through the diagonal Hessian, mitigates such issues and provides a more stable estimation framework.
>
> NPR-DDPM: This approach achieves strong likelihood results by learning the residual of the diagonal covariance using a pre-trained score network. However, its FID performance is weaker. We hypothesize that this is due to the inherent difficulty of learning the residual covariance, a more complex task than directly regressing the diagonal Hessian, as in our method.
>
> We hypothesize two reasons why our method achieves smaller estimation error:
>
> 1. Our approach directly regresses the diagonal covariance, which is simpler and more effective than estimating covariance through a loose lower bound.
> 2. Learning the diagonal covariance from the score helps reduce the high variance inherent in noisy data, compared to directly learning covariance from the noisy data. This potentially leads to improved performance.
>
> In summary, the improvement of our method may be attributed to the fact that regressing the diagonal Hessian directly from the pre-trained score is a more effective strategy for optimal covariance estimation compared to other baselines.

---

> > ### Author Response · Authors · 2024-11-19
> > **Response to Reviewer (Part 2)**
> >
> > > In Fig 3 (a) and (b), why the covariance error goes up for all methods when t is roughly in range [0, 400]?
> > >
> >
> > **ANSWER:** Sorry for the confusion. We plot the $L_2$ norm of the diagonal covariance across different $t$ in this [Figure](https://imgur.com/a/l2-norm-of-true-diagonal-covariance-in94eTc). The results show that the magnitude of the diagonal covariance increases as noise levels rise. Generally, score estimation and our diagonal Hessian estimation are very accurate at high noise levels but less accurate at low noise levels. Intuitively, at high noise levels, the covariance error remains small because the Hessian approximation is accurate. At low noise levels, despite the inaccuracy of the Hessian approximation, the covariance error is small due to the low magnitude of the true covariance. Consequently, significant covariance errors are observed only at intermediate noise levels, where the true covariance has a moderate magnitude, and the Hessian approximation is less precise. We will add this justification in the revised paper.

---

> > ### Comment · Reviewer_hfvy · 2024-11-21
> >
> > Thanks for the response!
> >
> > - stochastic and deterministic samplers
> >
> > I see, indeed it's possible to integrate the learned covariance into other samplers.The following could also be interesting.
> >
> > Actually whether outperforming the original EDM sampler or not is minor to me, I am curious whether the improved covariance also helps other types of diffusion models [1] / flow matching models [2] (these can be put into a general SDE/ODE framework, so integrating into SDE/ODE samplers is valuable to me), and how it helps?
> >
> > Let me cast the question this way: Most generative models is to gradually transport a simple source distribution A to a complex target distribution B. Previous works have shown that the choice of path/transition kernel, noise schedule, discretization, etc makes different transport schemes.
> > Then what's the function of getting optimal covariance estimate? Or say how does it improve the FID/NLL when taking fewer NFEs? For sure less cumulative error introduced, but how does it reduce the error and which error (discretization error, denoiser estimation error or something else?)? Is it something related to curvature of the process?
> >
> > - Thanks for the hypothesis. It's helpful to incorperate these into paper for easy understanding.
> >
> > [1] Karras, Tero, et al. "Elucidating the design space of diffusion-based generative models." Advances in neural information processing systems 35 (2022): 26565-26577.
> >
> > [2] Lipman, Yaron, et al. "Flow matching for generative modeling." arXiv preprint arXiv:2210.02747 (2022).

---

> ### Comment · Reviewer_hfvy · 2024-11-21
>
> Thanks for the clarification.
>
> I also read other reviews and your responses, that's very helpful and addresses some of my concerns which I was about to ask. Also thanks for the detailed replies, I really appriciate it. I've increased my score accordingly.
>
> My last comment may be less relavant to the paper, but I'm still interested to see your thoughts. I feel like that could be promising to explore.

---

> > ### Author Response · Authors · 2024-11-23
> > **Thanks for your insightful feedback**
> >
> > Thank you for taking the time to review our response and for providing additional constructive feedback. Below are our replies, and we are happy to discuss them further as your insights are highly valuable for future research.
> >
> > > Does the improved covariance also help other types of diffusion models (e.g., EDM) and flow models (e.g., flow matching)? Can the improved covariance be integrated into SDE/ODE samplers?
> > >
> >
> > **ANSWER:** Thank you for clarifying your concern. To address your question, we are pleased to provide further insights from two perspectives:
> >
> > - Can the covariance be learned for pretrained diffusion models or flow models of other types?
> >
> >     For diffusion models beyond DDPM, such as EDM, it is straightforward to learn the covariance given an EDM model. EDM employs a variance-exploding forward process $p(x_t|x_0) = \mathcal{N}(x_t; x_0, \sigma_t^2 I)$, which inherently defines an optimal covariance for the corresponding backward process. Therefore, we believe it is feasible to learn the covariance for general diffusion models whose forward processes can be discretized as Gaussian distributions.
> >
> >     For the flow models, it is not yet clear how to learn the covariance. However, we can provide a heuristic solution to inspire potential research in the future. Specifically, given a pretrained flow model with its velocity field, the corresponding score can be explicitly determined (as shown in Equation 9 of [1]), along with the associated SDE (as shown in Equation 4 of [1]). By discretizing this SDE into a Gaussian distribution, it becomes possible to learn the corresponding optimal covariance.
> >
> > - Can the improved covariance be integrated into SDE/ODE samplers?
> >
> >     As shown in [2, Appendix E], DDPM can be viewed as a specific instance of a discretized reverse SDE solver. We hypothesize that the denoising covariance can capture uncertainty arising from discretization errors, potentially linking it to probabilistic numerics methods [3, 4]. In [3], it is demonstrated that covariance estimates can inform adaptive step sizing: when uncertainty is low (indicated by small covariance), the solver can take larger steps safely, while higher uncertainty necessitates smaller steps to maintain accuracy. This insight may explain why our method has the potential to accelerate the sampling process. However, integrating the learned covariance into general SDE/ODE solvers and establishing an explicit connection to probabilistic numerics methods remains a complex challenge. We leave this as an intriguing direction for future research.
> >
> >
> > [1] Ma, Nanye, et al. "Sit: Exploring flow and diffusion-based generative models with scalable interpolant transformers." *arXiv preprint arXiv:2401.08740* (2024).
> >
> > [2] Song, Yang, et al. "Score-based generative modeling through stochastic differential equations." *arXiv preprint arXiv:2011.13456* (2020).
> >
> > [3] Schober, Michael, Simo Särkkä, and Philipp Hennig. "A probabilistic model for the numerical solution of initial value problems." *Statistics and Computing* 29.1 (2019): 99-122.
> >
> > [4] Hennig, Philipp, Michael A. Osborne, and Hans P. Kersting. *Probabilistic Numerics: Computation as Machine Learning*. Cambridge University Press, 2022.
> >
> > > How does learning covariance help to improve the FID/NLL when taking fewer NFEs. How does it reduce the error, and which type of error is being reduced — discretization error, denoiser estimation error, or something else?
> > >
> >
> > **ANSWER:** Thank you for the insightful question. For a fixed number of NFEs, the learned covariance can improve the tightness of the ELBO, leading to better NLL. Since the target covariance matched in our method corresponds to the optimal covariance under the KL divergence (see Theorem 2.3 in [5]), better covariance estimation results in a tighter ELBO (which equals the negative KL plus a constant).
> >
> > However, the relationship between increasing the number of time steps and the tightness of the ELBO remains an open problem (see Section 4.3.1.5 in [6]), which we identify as a promising direction for future work.
> >
> > [5] Zhang, Mingtian, et al. "Moment matching denoising Gibbs sampling." *Advances in Neural Information Processing Systems* 36 (2024).
> >
> > [6] Turner, Richard E., et al. "Denoising Diffusion Probabilistic Models in Six Simple Steps." *arXiv preprint arXiv:2402.04384* (2024).

---

### Official Review · Reviewer_tC1T · 2024-10-31

**Soundness:** 3
**Presentation:** 2
**Contribution:** 3
**Rating:** 8
**Confidence:** 3

**Summary:**

The paper proposes an approach to diffusion modelling in which the covariance matrix of the time-reversal of the "noising" process is set to a diagonal approximation of the correct form estimated from the approximation of the "score".

In order to do this a pair of neural networks are trained; one to estimate the score as usual and a second to estimate this diagonal matrix. The majority of parameters are shared between the two networks, allowing this to be accomplished with only a small additional computational cost. The properties of the objective function used to learn this covariance are explored to an extent in theoretical work.

Numerical results are presented for a toy model and for some more substantial image generation scenarios.

**Strengths:**

To the best of my knowledge this approach to approximating the covariance matrix is novel and seems well motivated.

The numerical results do appear to show that this approach outperforms a number of alternative strategies and the settings of the image generation setting do seem reasonably challenging.

It seems that the additional implementation and training requirements of this approach over and above those of a more basic diffusion generative model are modest and so this approach could be incorporated into implementations of such approaches at small cost (both human and computational) which means that even modest improvements are potentially valuable.

**Weaknesses:**

The paper doesn't seem to propose anything radically new; the introduction of sensible approximations of the covariance of the time reversal has been explored quite widely in recent years and so the contribution of this paper is the particular approach to doing that which, while well motivated and seemingly having good properties, does not seem particularly radical.

There are some odd choices made presentationally at times which to my mind make the paper a bit less readable and hence a bit less accessible. For example, the description of diffusion processes in Section 2.1 takes as a starting point a discrete time process; (1) is not a diffusion process although it is described as one. It might be arrived at as the time discretisation of a diffusion, which may in this case be exact, but that's not the same thing.
These are minor points and haven't substantially influenced my view of the paper, but this seems the best box to put this in. The paper does contain quite a number of typos, strangely phrased sentences and similar; careful proofreading would make it more readable. E.g.
* Abstract: "diagonal covariances" is an odd phrase; covariances are the off-diagonal elements of the matrix. "diagonal covariance matrix" or similar might be clearer.
* Theorem 2: "objetcive"
* Theorem 2: "The objective in Equation (11) upper bounded the grounded objective." Didn't immediately seem to make sense to me; I /thought/ the intention was probably something like "The objective in Equation (11) upper bounds the base objective." or something of that sort, but it's not completely clear what the intended base objective was. It turns out that aside from the odd tense and phrasing, the statement of this theorem is using terms which are defined in the supplementary material!

The main evidence presented in support of the proposed approach is empirical and this evidence isn't completely overwhelming. E.g. from Table 3 it' seems that this method doesn't actually dominate other approaches in the literature although it often does well and doesn't seem to be outperformed by much when it is beaten. Of course, methods can be interesting without beating the state of the art uniformly, and there is the interesting suggestion that when other methods are outperforming this approach in terms of NLL they do less well in FID terms. That, of course, raises the question "why is this and what does it actually tell us?"

**Questions:**

Line 36: in what sense is this a "posterior" covariance? It seems from later in the paper that the covariance of the semigroup of the time reversal of the noising process is what's intended; I don't see that there's anything Bayesian going on.

Line 199: "In practice, we found M = 1 works well," raises some obvious questions:
1. How did you find this?
2. What does "works well" mean and how does increasing M to larger numbers change behaviour? A single sample provides an unbiased estimate, but its variance will clearly be large; generally one would expect performance to increase with larger M and so the question is how does the performance/cost tradeoff behave and this offhand comment doesn't give the reader much information which is a shame if you've done a proper study of this.

Have you done any comparison with the full covariance estimation approach of Zhang et al. (2024b)? While I appreciate it may sometimes be prohibitively costly it would have been nice to see what price is paid for making the diagonal approximation in at least as toy scenario. It seems as though strong correlation between coordinates in the true distribution might be reflected by strong correlation in the time reversal of the noising process, for example, and in those settings approximating it with a diagonal covariance although quite possibly still better than a naive choice might suffer relative to a more flexible approach whereas in settings in which there is not such strong dependency modelling it in the covariance matrix might not be particularly important.

---

> ### Author Response · Authors · 2024-11-19
> **Response to Reviewer (Part 1)**
>
> We appreciate the reviewer's constructive feedback on our paper. Regarding the weaknesses pointed out, we would like to provide the following clarifications:
>
> > The paper doesn't seem to propose anything radically new; the introduction of sensible approximations of the covariance of the time reversal has been explored quite widely in recent years and so the contribution of this paper is the particular approach to doing that which, while well motivated and seemingly having good properties, does not seem particularly radical.
> >
>
> **ANSWER:**
>
> We appreciate the reviewer’s acknowledgement of the motivation behind our method and its promising properties. We would like to emphasize that our contribution lies in introducing a new learning objective aimed at reducing the approximation error in diagonal covariance estimation. This innovation enables our method to achieve an optimal trade-off between FID and likelihood metrics with fewer NFEs, which has been a persistent challenge for existing methods.
>
> For example, some approaches prioritize better FID at the expense of likelihood (e.g., SN-DDPM), while others improve likelihood but compromise FID (e.g., NPR-DDPM). Our method addresses this trade-off effectively, offering a balanced improvement in both metrics.
>
> To explicitly highlight this advantage, we have included a new trade-off plot that varies the number of network evaluations on CIFAR-10 (CS). Please refer to this [Figure](https://imgur.com/a/fid-nll-trade-off-8ctJg8S) for illustration. We will add this plot to the revised version to emphasize the benefits of our approach.
>
> Moreover, we attribute the main insights of our research to the observation that better covariance approximation can lead to improving performance. We hope this insight can inspire future research on more advanced covariance approximation techniques, such as low-rank and block-diagonal approximation.
>
> > There are some odd choices made presentationally at times which to my mind make the paper a bit less readable and hence a bit less accessible.
> >
>
> **ANSWER:** Thank you for pointing these out. We have revised them based on your suggestions. We greatly appreciate these suggestions that make the paper more readable.

---

> > ### Author Response · Authors · 2024-11-19
> > **Response to Reviewer (Part 2)**
> >
> > > The main evidence presented in support of the proposed approach is empirical and this evidence isn't completely overwhelming. E.g. from Table 3 it' seems that this method doesn't actually dominate other approaches in the literature although it often does well and doesn't seem to be outperformed by much when it is beaten. Of course, methods can be interesting without beating the state of the art uniformly, and there is the interesting suggestion that when other methods are outperforming this approach in terms of NLL they do less well in FID terms. That, of course, raises the question "why is this and what does it actually tell us?”
> > >
> >
> > **ANSWER:** Thank you for your insightful question. We emphasize that its key advantage is our method can lead to lower estimation of the diagonal covariance which can further lead to better FID-likelihood trade-off with fewer steps. Below, we provide a detailed hypothesis to explain why learning covariance via the OCM objective outperforms baseline methods across all noise levels.
> >
> > I-DDPM: This method jointly learns the mean and covariance by maximizing the evidence lower bound (ELBO) of the likelihood. We hypothesize that this joint optimization introduces challenges because the covariance estimation depends heavily on the mean. When the mean is suboptimal during training, the ELBO becomes overly loose, resulting in an additional bias in the covariance learning. In contrast, our method simplifies this process by directly regressing the target covariance, which avoids the complexities of optimizing a loosely bound and leads to a less biased learning procedure.
> >
> > SN-DDPM:  This method estimates diagonal covariance directly from noisy empirical data. While it improves FID in some cases, it struggles with likelihood estimation. A key issue here is the quadratic term in their model (as discussed in Equation 24 of their paper), which amplifies estimation errors. This effect is particularly pronounced at low noise levels, where score estimation is less accurate. Our method, by directly targeting covariance estimation through the diagonal Hessian, mitigates such issues and provides a more stable estimation framework.
> >
> > NPR-DDPM: This approach achieves strong likelihood results by learning the residual of the diagonal covariance using a pre-trained score network. However, its FID performance is weaker. We hypothesize that this is due to the inherent difficulty of learning the residual covariance, a more complex task than directly regressing the diagonal Hessian, as in our method.
> >
> > We hypothesize two reasons why our method achieves smaller estimation error:
> >
> > 1. Our approach directly regresses the diagonal covariance, which is simpler and more effective than estimating covariance through a loose lower bound.
> > 2. Learning the diagonal covariance from the score helps reduce the high variance inherent in noisy data, compared to directly learning covariance from the noisy data. This potentially leads to improved performance.
> >
> > In summary, the improvement of our method may be attributed to the fact that regressing the diagonal Hessian directly from the pre-trained score is a more effective strategy for optimal covariance estimation compared to other baselines.
> >
> > > Line 36: in what sense is this a "posterior" covariance? It seems from later in the paper that the covariance of the semigroup of the time reversal of the noising process is what's intended; I don't see that there's anything Bayesian going on.
> > >
> >
> > **ANSWER:** Thank you for pointing this out. We have revised it to be “the covariance of denoising distributions”.

---

> > > ### Author Response · Authors · 2024-11-19
> > > **Response to Reviewer (Part 3)**
> > >
> > > > Line 199: "In practice, we found M = 1 works well," raises some obvious questions:
> > > >
> > > > 1. How did you find this?
> > > > 2. What does "works well" mean and how does increasing M to larger numbers change behaviour? A single sample provides an unbiased estimate, but its variance will clearly be large; generally one would expect performance to increase with larger M and so the question is how does the performance/cost tradeoff behave and this offhand comment doesn't give the reader much information which is a shame if you've done a proper study of this.
> > >
> > > **ANSWER:** Thank you for making us aware of this. The intuition behind this argument is two-fold: Firstly, the training loss defined in Equation 11 is unbiased, ensuring that even with a single Rademacher sample, it can accurately recover the true diagonal covariance under optimal training. Secondly, It is noteworthy that a similar approach is also employed in denoising score matching. Specifically, given the denoising score identity
> > >
> > > $$
> > > \nabla\_{x_t} \log p(x_t) = \mathbb{E}\_{p(x_0 | x_t)}[\nabla\_{x_t} \log p(x_t | x_0)],
> > > $$
> > >
> > > we can learn a neural network $s_\theta(x_t)$ to approximate $\nabla_{x_t} \log p(x_t)$ by minimising the following loss
> > >
> > > $$
> > > \mathcal{L}(\theta) = \mathbb{E}\_{p(x_t)} \lVert s_{\theta (x_t)} - \mathbb{E}\_{p(x_0 | x_t)}[ \nabla\_{x_t} \log p(x_t | x_0)] \rVert_2^2,
> > > $$
> > >
> > > in which the inner expectation can be estimated using multiple Monte Carlo samples to reduce variance; however, this method is biased and impractical. Similar to Theorem 2 in our paper, this loss function can be reformulated into the following form:
> > >
> > > $$
> > > \mathcal{L}(\theta) = \mathbb{E}\_{p(x_t)p(x_0|x_t)} \lVert s\_{\theta (x_t)} - \nabla\_{x_t} \log p(x_t | x_0) \rVert_2^2 + c.
> > > $$
> > >
> > > This objective, commonly referred to as denoising score matching, is unbiased and works well with a single Monte Carlo sample $x_0 \sim p(x_0 | x_t)$.
> > >
> > > To validate our argument, we include an ablation study examining the effect of varying the number of Rademacher samples $M$. Specifically, we conduct experiments on CIFAR10 (CS) using the proposed OCM-DDPM method and evaluate performance based on FID and NLL. The empirical results, presented in the following tables, indicate that different values of $M$ yield similar performance. This is practically desirable as, in practice, setting $M$=1 allows for more efficient training.
> > >
> > > FID results:
> > >
> > > |  | K=10 | K=25 | K=50 | K=100 |
> > > | --- | --- | --- | --- | --- |
> > > | M=1 | 14.32 | 5.54 | 4.10 | 3.84  |
> > > | M=3 | 14.18 | 5.51 | 4.11 | 3.82 |
> > > | M=5 | 14.17 | 5.51 | 4.11 | 3.82 |
> > > | M=10 | 14.16 | 5.51 | 4.11 | 3.82 |
> > >
> > > NLL results:
> > >
> > > |  | K=10 | K=25 | K=50 | K=100 |
> > > | --- | --- | --- | --- | --- |
> > > | M=1 | 4.99 | 4.34 | 3.99 | 3.76 |
> > > | M=3 | 4.99 | 4.34 | 4.00 | 3.76 |
> > > | M=5 | 4.99 | 4.34 | 4.00 | 3.76 |
> > > | M=10 | 4.99 | 4.34 | 4.00 | 3.76 |
> > >
> > > We will include the discussion of using different numbers of Rademacher samples in the revision.

---

> > > > ### Author Response · Authors · 2024-11-19
> > > > **Response to Reviewer (Part 4)**
> > > >
> > > > > Have you done any comparison with the full covariance estimation approach of Zhang et al. (2024b)? While I appreciate it may sometimes be prohibitively costly it would have been nice to see what price is paid for making the diagonal approximation in at least as toy scenario. It seems as though strong correlation between coordinates in the true distribution might be reflected by strong correlation in the time reversal of the noising process, for example, and in those settings approximating it with a diagonal covariance although quite possibly still better than a naive choice might suffer relative to a more flexible approach whereas in settings in which there is not such strong dependency modelling it in the covariance matrix might not be particularly important.
> > > > >
> > > >
> > > > **ANSWER:** Thank you for raising this concern. As you mentioned, using full covariance estimation in image modelling is prohibitively expensive. For instance, computing the Jacobian on CIFAR10 data ($3\times32\times32$) takes approximately 9 minutes on an RTX 3090 GPU with a batch size of 64. This makes it computationally infeasible, as generating 50K images with 10 steps and calculating the FID would require around 1,171 hours.
> > > >
> > > > Alternatively, as you suggested, we can apply full covariance in our toy demonstration in Section 5.1. Specifically, we propose including a baseline that performs sampling using the true full covariance (please refer to the anonymous link for the [DDPM](https://imgur.com/a/ddpm-mmd-v-s-steps-Qfx5MRc) and [DDIM](https://imgur.com/a/ddim-mmd-v-s-steps-bkXOCzg) results). The results indicate that the diagonal approximation and full covariance yield similar performance in this two-dimensional mixture of Gaussians (MoG) toy problem. This is likely because the Gaussian components in the MoG have isotropic covariance structures.
> > > >
> > > > However, we believe that employing more advanced covariance approximation methods could improve performance in high-dimensional problems, where local covariance structures are more prevalent. For instance, a low-rank approximation could be explored by modelling a neural network $L_\theta (x_t)$ and learning it by minimizing the following loss:
> > > >
> > > > $$
> > > > ||L(x_t)L(x_t)^T+\sigma^2I-\nabla^2_{x_t} \log p_t(x_t))||_2^2
> > > > $$
> > > >
> > > > However, this approach requires explicit computation of the Hessian, which is computationally expensive in practice. Additionally, the projection method used in our paper can only guarantee the convergence of the diagonal terms and cannot be applied to learn the off-diagonal structure. Therefore, developing efficient methods for applying low-rank approximation in optimal covariance matching remains a challenging but promising direction for future work. We will include this discussion in the revised version.

---

> > > > > ### Comment · Reviewer_tC1T · 2024-11-20
> > > > > **Thanks...**
> > > > >
> > > > > for the very detailed and interesting response.
> > > > >
> > > > > The consideration of the tradeoff between NLL and FID is interesting, and the additional experiments showing the impact of M are interesting.
> > > > >
> > > > > My reservations have to an extent been addressed and I will increase my score, although I remain of the opinion that an illustration on a low-dimensional toy problem showing what impact neglecting off-diagonal elements of the covariance might have would have been a valuable addition.

---

> > > > > > ### Author Response · Authors · 2024-11-23
> > > > > > **Thanks for your constructive feedback**
> > > > > >
> > > > > > Apologies for the delayed response, and thank you for your positive feedback on our work! We are pleased to inform you that we have added a new experiment to demonstrate the impact of using full covariance. In this new experiment, we use a mixture of 40 Gaussians (see [this figure](https://imgur.com/a/40-mog-RiSq2pU) for the visualisation of data and density). Unlike the MoG shown in Figure 3 of the paper, this new MoG does not have symmetric means for its components, resulting in a non-diagonal covariance structure. The results (please refer to the [DDPM](https://imgur.com/a/ddpm-mmd-v-s-steps-aVdKWMR) and [DDIM](https://imgur.com/a/ddim-mmd-v-s-steps-LRqNTMz) results) show that learning a diagonal covariance improves performance compared to using a fixed isotropic covariance. Additionally, using the true full covariance leads to even better results than the diagonal covariance. We hope that these new findings provide further insights and inspire future research to explore more advanced approximations of the full covariance, such as low-rank or block-diagonal approximations.

---

### Official Review · Reviewer_mSjs · 2024-11-03

**Soundness:** 3
**Presentation:** 3
**Contribution:** 3
**Rating:** 8
**Confidence:** 4

**Summary:**

The paper propose to build on existing moment matching diffusion progress and focus on the estimation of the optimal covariance. They propose to improve over previous methods for estimating the diagonal of covariance with an additional network that amortizes the cost of computing explicitly the covariance during sampling. They show that their method is challenging with selected comparisons.

**Strengths:**

Thank you for your work!

- The paper demonstrates the application of OCM across different models (classical and latent diffusion), making the approach adaptable for a range of diffusion-based applications.

- The paper target efficient diagonal estimation to reach efficient sampling (leading to reduced number of step and more accurate samples).

- The proposed method is timely as it is based on recent covariance matching methods. It correctly try to improve efficiency over existing estimation techniques by amortizing the covariance estimation with an additional neural network. It still bases its estimation on learned diffusion model, making it comparable to previous ideas.

- It looks like very reasonable overhead for the Hutchinson estimator (M = 1 as noted in line 199) provide good performance which is in favor of the method that seeks efficiency.

- They propose an unbiased objective for diagonal covariance matching.

**Weaknesses:**

In general, the paper is good. However the results miss visualization (in the core text). A toy example would be welcome for visualization as well and easy results interpretation/comparison. The performances reported do not exhibit high improvement over other methods. Although it shows to be challenging! Maybe other performances should be reported to strengthen the defense of this method over others (the authors often point efficiency as a main goal, but no convincing observations in this favor

**General Comments**

- The title is a bit misleading. You do not make optimal covariance matching, only diagonal matching. I also found the choice of diagonal only matching a little bit skipped. I do not say that it's a bad approximation, but it has its limitations. You should motivate more 'why' diagonal is already a good approximation (more than the complexity motivation).

- Figure 1 is not really relevant as so. Ok the number of Rademacher samples improves the diagonal structure, but we don't know the target. Could you estimate it for comparison? Or doing the same for a toy problem with known covariance? Can you quantify or give an idea of information loss induced by only estimating the diagonal (It depends on the problem of course)? Figure 1 do not provide more information than Appendix C4. Maybe the latter can be replaced by something that focus on your method. The number of Rademacher samples do not seems to be the main point of your method.

- Concerning Rademacher samples, it would be nice to see a metric (like FID and IS) evolving with M (the number of samples) to strengthen your claim line 199.

- Echoing with the previous point, it would be nice to show more extensive comparison with other covariance matching methods. Either that target full covariance or more costly diagonal estimation. Insist more on the advantage of your estimator concerning the estimation speed/accuracy ratio. In the same trend as table 4. Maybe you can have a small toy problem (MoG, ...) for which it is better to interpret results (and visualize them) + having access to GT covariance.

- I'm a little bit worried about the 'diagonal' only estimation. What if you have strong local covariance? Could be good to discuss, give hints about potential failure of your method in such setting. I take the example of [1] where they have to define a new sampling strategy as existing ones do not capture correctly covariance (or assumptions are too strong) leading to poor sample accuracy. This method also provides moment matching sampling, efficient by using vjp in spite of modelling the full covariance explicitly. Could you compare to similar method? For example, how do you compare to low-rank estimation which are also less demanding but can model non-diagonal elements?

- In the same idea as [1], current diffusion model and sampling strategies hardly comes without posterior sampling. It would be nice to study the benefit of your estimation for posterior problem. I think that is needed for a sampling method.

- In equation (10), you still need $H(\tilde{x})$ which can be large as you mention. Do you model it explicitly for training the covariance network or do you model it implicitly using jvp/vjp benefits? This should be discussed. How does it slow down training compared to classical denoising-score matching or other methods?

- Can the introduction of an additional network (meaning additional approximation) lead to cumulative errors? This is not discussed at all, but the network not being well aligned with the covariance of the trained diffusion will surely lead to poor performances even if the diffusion model was good.

**Spotted typos**

- Figure 1: The $K$ corresponds to $M$ in the main text, no? Could be changed for consistency.

- Line 195 and 812: ~Validity of OCM objetcive~ -> Validity of OCM objective.

- Line 309 and 427 ~COM-DDPM~ and ~OCM-DPM~ -> OCM-DDPM, no?

- Appendix A.2: ~Vadality~ -> Validity.

I will be pleased to increase my score once my questions and concerns are answered, discussed and addressed.

**[1] Learning Diffusion Priors from Observations by Expectation Maximization - 2024**
I took this one as example, you can recursively check references which are relevant for this paper also.

**Questions:**

See Weaknesses.

---

> ### Author Response · Authors · 2024-11-19
> **Response to Reviewer (Part 1)**
>
> Thanks a lot for your valuable comments. Your suggestions are very helpful in further improving the work, and we are refining the manuscript accordingly.
>
> > In general, the paper is good. However the results miss visualization (in the core text). A toy example would be welcome for visualization as well and easy results interpretation/comparison. The performances reported do not exhibit high improvement over other methods. Although it shows to be challenging! Maybe other performances should be reported to strengthen the defense of this method over others (the authors often point efficiency as a main goal, but no convincing observations in this favor
> >
>
> **ANSWER:** Thanks for your valuable comments. We recognize the importance of clear visual demonstrations.  In Section 5.1 and Figure 3, we have already provided a toy example using a mixture of Gaussians, where we compare covariance estimation accuracy and DDPM/DDIM sampling efficacy and show that our method archives lower estimation error and better sampling performance. However, we understand this could be elaborated further. In the revised version, we will include additional visualization to enhance clarity and facilitate result interpretation and comparison.
>
> While our method may not outperform all strong baselines in every setting, we emphasize that its key advantage is achieving the best FID-likelihood trade-off with fewer steps. Following your suggestion, we have included an additional visualization on CIFAR10 (CS) to better highlight this benefit (please refer to [this figure](https://imgur.com/a/fid-nll-trade-off-8ctJg8S) for details). Furthermore, the central insight of our paper is that improved covariance estimation can lead to better performance in both sample quality (FID) and density estimation (likelihood). We hope this insight will inspire future research. For example, more advanced covariance approximation methods, such as low-rank approximations, as suggested by the reviewer, represent a promising direction for future work.
>
> > The title is a bit misleading. You do not make optimal covariance matching, only diagonal matching. I also found the choice of diagonal only matching a little bit skipped. I do not say that it's a bad approximation, but it has its limitations. You should motivate more 'why' diagonal is already a good approximation (more than the complexity motivation).
> >
>
> **ANSWER:** Thanks for raising this point. In the revision, we will clarify the use of diagonal covariance more clearly in the title/abstraction to prevent potential misinterpretation. Our choice to match the diagonal covariance is primarily driven by considerations of complexity and scalability, aligning it with the architectures commonly employed in mainstream structures such as iDDPM.  Our primary goal is to demonstrate that the proposed objective of learning the state-dependent diagonal approximation is better than the state-independent isotropic covariance or other diagonal covariance learning methods, which are common techniques used in diffusion sampling. Additionally, we added an experiment on comparing full covariance and the learned diagonal in the toy MoG setting (please check the [DDPM](https://imgur.com/a/ddpm-mmd-v-s-steps-Qfx5MRc) and [DDIM](https://imgur.com/a/ddim-mmd-v-s-steps-bkXOCzg) results), where we found that the performance of the true full covariance and learned diagonal are similar.
>
> However, we do agree with the reviewer that the covariance structure will be more important in the high dimensional case, where the sparse approximation like low-rank approximation $L_\theta (x_t) L_\theta (x_t)^T$ or block-diagonal approximation can give more flexible structure information and potentially improve the effectiveness of the optimal covariance approximation. However, how to find a scalable loss to efficiently learn $L_\theta$ or the block-diagonal for large dimensional data is a challenging task which could not be easily integrated into the proposed loss function (see more discussion in the comparison to the setting in the inverse problem below). We thus only focus on how to improve the diagonal covariance estimation accuracy, leaving more flexible approximation as a promising direction for future research. We want to thank the reviewer again for pointing out this exciting future direction, we will add a relevant discussion in our revised paper.

---

> > ### Comment · Reviewer_mSjs · 2024-11-21
> > **Thanks (part 1)**
> >
> > Thanks for your answers! It's very complete. I will detail my remaining concerns in every response. However, I like your efforts and the clarifications you made. I'll then raise my score accordingly.
> >
> > 1) The new figure is nice to emphasize the trade-off between efficiency and exactness. Thanks.
> >
> > 2) Could you put the y-axis in log scale for both DDIM and DDPM MMD on MoG? How would you explain the higher MMD for the exact covariance at $steps=2$? Do not hesitate to correctly detail your MoG experiment as the latter could be easy (or not) to solve with a diagonal covariance. It would be interesting to see how it evolves with the strength of off-diagonal elements (do not add that, you already have way enough! Discuss it if you can). Is the MoG experiment the same as Fig. 3? If yes, 2-dimensional is a very small problem but fine.

---

> > > ### Author Response · Authors · 2024-11-23
> > > **Follow-up Response (Part 1)**
> > >
> > > > Could you put the y-axis in log scale for both DDIM and DDPM MMD on MoG? How would you explain the higher MMD for the exact covariance at steps=2? Do not hesitate to correctly detail your MoG experiment as the latter could be easy (or not) to solve with a diagonal covariance. It would be interesting to see how it evolves with the strength of off-diagonal elements (do not add that, you already have way enough! Discuss it if you can). Is the MoG experiment the same as Fig. 3? If yes, 2-dimensional is a very small problem but fine.
> > > >
> > >
> > > **ANSWER:** Thanks for your further comments. Here are our responses.
> > >
> > > - Could you put the y-axis in log scale for both DDIM and DDPM MMD on MoG?
> > >
> > >     Thank you for the suggestion. However, we find it to be noisy since the MMD approaches  0 as $t$ increases, and the logarithmic scale amplifies the difference, potentially introducing noise that could distract the reader. Here is [the log-scale version](https://imgur.com/a/ddpm-log-mmd-v-s-steps-QWr0p6j) of Figure 3(c).
> > >
> > > - How would you explain the higher MMD for the exact covariance at steps=2 It would be interesting to see how it evolves with the strength of off-diagonal elements.
> > >
> > >     Thank you for bringing this to our attention. In the previous figure, we used the learned mean and the true covariance. The poor MMD result of GT-COV at steps=2 was likely due to the incompatibility between the learned mean and the true covariance. To avoid the potential confusion, we have now updated the figure to use the true mean and true covariance for GT COV, while retaining the learned mean for the other baselines (please refer to the [DDPM results](https://imgur.com/a/ddpm-mmd-v-s-steps-PlQms0W) and [DDIM results](https://imgur.com/a/ddim-mmd-v-s-steps-fDS0TRZ)). As shown in the updated figure, using the true mean and true covariance leads to best performance.
> > >
> > >     We found in this simple mixture of nine Gaussian with symmetric components, the true covariance is dominated by the diagonal entries. This is consistent with the demonstrated results that diagonal covariance and full covariance has similar performance. To further  study the benefit of including the non-diagonal term, we have added a more complex toy example using the mixture of 40 Gaussians (see [this figure](https://imgur.com/a/40-mog-RiSq2pU) for the visualisation of data and density). Unlike the MoG shown in Figure 3 of the paper, this new MoG does not have symmetric means for its components, resulting in a non-diagonal covariance structure. The results (please refer to the [DDPM](https://imgur.com/a/ddpm-mmd-v-s-steps-aVdKWMR) and [DDIM](https://imgur.com/a/ddim-mmd-v-s-steps-LRqNTMz) results) show that using the true full covariance leads to even better results compared to the true diagonal covariance. Our method ranks third, demonstrating improved performance compared to using a fixed isotropic covariance or other diagonal learning approaches. We are highly encouraged by this outcome and sincerely grateful to the reviewer for this insightful suggestion. We believe this represents a promising future direction for developing more efficient methods to learn the off-diagonal terms.
> > >
> > >
> > > - Is the MoG experiment the same as Fig. 3? If yes, 2-dimensional is a very small problem but fine.
> > >
> > >     Yes, the MoG experiment is the same as Fig.3. It is a two-dimensional toy example.

---

> ### Author Response · Authors · 2024-11-19
> **Response to Reviewer (Part 2)**
>
> > Concerning Rademacher samples, it would be nice to see a metric (like FID and IS) evolving with M (the number of samples) to strengthen your claim line 199.
> >
>
> **ANSWER:** Thanks for the great suggestion. Notably, the training loss defined in Equation 11 is unbiased, meaning that even with a single Rademacher sample, it can accurately recover the true diagonal covariance under optimal training.
>
> To validate our argument, we include an ablation study examining the effect of varying the number of Rademacher samples $M$. Specifically, we conduct experiments on CIFAR10 (CS) using the proposed OCM-DDPM method and evaluate performance based on FID and NLL. The empirical results, presented in the following tables, indicate that a larger M (e.g. M=3) can give a small improvement in FID. This improvement is likely due to the reduced gradient estimation variance during training with a larger M. However, in most cases, different values of $M$ yield consistent performance.
>
> This is practically desirable as, in practice, setting $M$=1 allows for efficient training while maintaining strong performance, While larger M may offer marginal gains in FID, the trade-off between computational efficiency and performance improvement will be further discussed in our revised version. We sincerely thank the reviewer for this insightful suggestion!
>
> FID results:
>
> |  | K=10 | K=25 | K=50 | K=100 |
> | --- | --- | --- | --- | --- |
> | M=1 | 14.32 | 5.54 | 4.10 | 3.84  |
> | M=3 | 14.18 | 5.51 | 4.11 | 3.82 |
> | M=5 | 14.17 | 5.51 | 4.11 | 3.82 |
> | M=10 | 14.16 | 5.51 | 4.11 | 3.82 |
>
> NLL results:
>
> |  | K=10 | K=25 | K=50 | K=100 |
> | --- | --- | --- | --- | --- |
> | M=1 | 4.99 | 4.34 | 3.99 | 3.76 |
> | M=3 | 4.99 | 4.34 | 4.00 | 3.76 |
> | M=5 | 4.99 | 4.34 | 4.00 | 3.76 |
> | M=10 | 4.99 | 4.34 | 4.00 | 3.76 |
>
> Furthermore, we are happy to provide additional insights into the OCM objective. It is noteworthy that a similar approach is also employed in denoising score matching. Specifically, given the denoising score identity
>
> $$
> \nabla_{x_t} \log p(x_t) = \mathbb{E}_{p(x_0 | x_t)} [\nabla\_{x_t} \log p(x_t | x_0)],
> $$
>
> we can learn a neural network $s_\theta(x_t)$ to approximate $\nabla\_{x_t} \log p(x_t)$ by minimising the following loss
>
> $$
> \mathcal{L}(\theta) = \mathbb{E}\_{p(x_t)} \lVert s\_{\theta (x_t)} - \mathbb{E}\_{p (x_0 | x_t)}[ \nabla\_{x_t} \log p(x_t | x_0)] \rVert_2^2,
> $$
>
> in which the inner expectation can be estimated using a large number of Monte Carlo samples to reduce variance; however, this method is biased and impractical. Similar to Theorem 2 in our paper, this loss function can be reformulated into the following form:
>
> $$
> \mathcal{L}(\theta) = \mathbb{E}\_{p(x_t)p(x_0|x_t)} \lVert s_{\theta (x_t)} - \nabla\_{x_t} \log p(x_t | x_0) \rVert_2^2 + c.
> $$
>
> This objective, commonly referred to as denoising score matching, is unbiased and works well with a single Monte Carlo sample $x_0 \sim p(x_0 | x_t)$.
>
> > it would be nice to show more extensive comparison with other covariance matching methods. Either that target full covariance or more costly diagonal estimation. Insist more on the advantage of your estimator concerning the estimation speed/accuracy ratio. In the same trend as table 4. Maybe you can have a small toy problem (MoG, ...) for which it is better to interpret results (and visualize them) + having access to GT covariance.
> >
>
> **ANSWER:** Thanks for the great suggestion. We compare our method to other covariance matching approaches using a toy problem in Figure 3, where the data distribution is a two-dimensional mixture of Gaussians (MoG). In this scenario, we have access to the true covariance matrix. Figures 3(a) and 3(b) illustrate the $L_2$ error of the estimated diagonal covariance across different noise levels, demonstrating that our method achieves the lowest error for all $t$ values.
>
> To demonstrate the sampling efficacy of the learned covariance, we also perform DDPM and DDIM sampling for the MoG problem. As shown in Figures 3(c) and 3(d), our method outperforms others with fewer generation steps. Additionally, we consider including a baseline that samples using the true full covariance (please click the anonymous link to see the [DDPM](https://imgur.com/a/ddpm-mmd-v-s-steps-Qfx5MRc) and [DDIM](https://imgur.com/a/ddim-mmd-v-s-steps-bkXOCzg) results). The results show that our method achieves performance comparable to the full covariance. However, it is worth noting that while the diagonal approximation and full covariance yield similar performance in this two-dimensional toy problem, we believe that approximating the full covariance would achieve better performance in high-dimensional problems, albeit at the cost of increased computational complexity.

---

> > ### Author Response · Authors · 2024-11-19
> > **Response to Reviewer (Part 3)**
> >
> > > I'm a little bit worried about the 'diagonal' only estimation. What if you have strong local covariance? Could be good to discuss, give hints about potential failure of your method in such setting. I take the example of [1] where they have to define a new sampling strategy as existing ones do not capture correctly covariance (or assumptions are too strong) leading to poor sample accuracy. This method also provides moment matching sampling, efficient by using vjp in spite of modelling the full covariance explicitly. Could you compare to similar method? For example, how do you compare to low-rank estimation which are also less demanding but can model non-diagonal elements?
> > >
> >
> > **ANSWER:** Thank you for the recommendation of the related paper. We note that the goal of paper [1] is to approximate the score $\nabla_{x_t} \log p(y|x_t)$, which involves solving a linear system $c = A^{-1}b$ using the conjugate gradient method. This approach implicitly accounts for the full covariance. In our case, however, we aim to sample from $p(x_{t-1}|x_t) = \mathcal{N}(\mu, \Sigma)$, which requires a Cholesky decomposition to compute $\Sigma^{\frac{1}{2}}\epsilon$ for sampling. Unfortunately, this cannot be solved using the conjugate gradient method with JVP. Thus, we believe these represent two distinct problems, each with its own unique challenges. We will cite it and include a discussion in the revised version.
> >
> > We fully agree with the reviewer that incorporating full-rank, low-rank, or block-diagonal covariance during sampling would be advantageous, as strong local covariance often exists in high-dimensional data. However, learning such a flexible structure while ensuring **training tractability**, where parameters and memory scale linearly with the data dimensions, remains a challenging problem.  For example, for the low-rank approximation, naively minimizing $||L(x_t)L(x_t)^T+\sigma^2I-\nabla^2_{x_t} \log p_t(x_t))||_2^2$ , where $L(x_t)$ is a low-rank matrix, becomes computationally prohibitive for large-dimensional settings data, and the projection method used in our paper can only grantee the converges of the diagonal term. We leave them as a promising research direction.
> >
> > Moreover, the current focus of our paper is to improve the estimation of the diagonal terms and demonstrate how better estimation accuracy can lead to performance improvements. This outcome reinforces confidence in exploring more structured covariance approaches in the future. We will add this discussion in our revised version. Thank you for the inspiring comment.
> >
> > [1] Rozet, François, et al. "Learning Diffusion Priors from Observations by Expectation Maximization." *arXiv preprint arXiv:2405.13712* (2024).
> >
> > > In the same idea as [1], current diffusion model and sampling strategies hardly comes without posterior sampling. It would be nice to study the benefit of your estimation for posterior problem. I think that is needed for a sampling method.
> > >
> >
> > **ANSWER:** Thank you again for the valuable reference to [1]. Our method can benefit the inverse problems, particularly when the goal is to improve sampling efficiency in the inference time. Specifically, the term $\mathbb{V}[x|x_t]$ (see equation 20 in [1]) can be replaced with our approximated diagonal covariance, and we can compute the $\nabla_{x_t}\log q(y|x_t)$ with only one NFE. This approach is significantly faster compared to using a conjugate gradient, which requires multiple NFEs.
> >
> > However, as we mentioned in the previous comment, our diagonal approximation may yield lower flexibility than the conjugate gradient method which considers the full covariance structure in this setting, so there is an accuracy/efficiency trade-off in the inverse problem setting when choosing between our method and the conjugate gradient method in [1].
> >
> > We will include a discussion on applying our method to inverse problems in the revision and plan to explore this further in future work.

---

> > > ### Comment · Reviewer_mSjs · 2024-11-21
> > > **Thanks (part 3)**
> > >
> > > 1) I do not ask you to use the conjugate gradient method. I was just wondering how you would compare to such a method (among others) that has another covariance estimation. I cited this paper as it resonates with several of my questions (what about posterior sampling with your method, what about other covariance assumptions, ...).
> > >
> > > However, as you pointed in the second answer, you could replace $\mathbb{V}[x\mid x_t]$ with your estimation and still sample from the marginal, without conditioning. I think that what comes out from this discussion is that, your method is twofold. From one hand you have the diagonal approximation and the unbiased loss, on the other hand you have a 'new' sampling method based on this new estimation. Both sides can be discussed separately. Typically, questions about time complexity of your method compared to others, posterior problems, ... are more 'sampling algorithm' questions.

---

> > > > ### Author Response · Authors · 2024-11-23
> > > > **Follow-up Response (Part 3)**
> > > >
> > > > > I do not ask you to use the conjugate gradient method. I was just wondering how you would compare to such a method (among others) that has another covariance estimation. I cited this paper as it resonates with several of my questions (what about posterior sampling with your method, what about other covariance assumptions, ...).
> > > > However, as you pointed in the second answer, you could replace with your estimation and still sample from the marginal, without conditioning. I think that what comes out from this discussion is that, your method is twofold. From one hand you have the diagonal approximation and the unbiased loss, on the other hand you have a 'new' sampling method based on this new estimation. Both sides can be discussed separately. Typically, questions about time complexity of your method compared to others, posterior problems, ... are more 'sampling algorithm' questions.
> > > > >
> > > >
> > > > **ANSWER:** Thank you for your additional comments; they are very insightful. Although the method proposed in that paper cannot be directly applied to the general sampling problem in diffusion (as it relies on the JVP, which is not applicable when sampling from a Gaussian with full covariance), we believe that applying our method to solve the inverse problem is a promising direction for future work. We appreciate the reviewer for providing such valuable suggestions.
> > > >
> > > > > What do you mean per iteration for your table? It is per training iteration? It's expected and the time complexity is reasonably above, but thanks to explicit that! Would also be nice to see something similar but at inference (where you do not need jvp anymore).
> > > > >
> > > >
> > > > **ANSWER:** Yes, it is per training iteration. We have included an inference time comparison in Table 4. The multiplication coefficient in the table represents the time cost per single sampling timestep. As shown, the additional sampling time cost is negligible because, as you mentioned, we do not need JVP during sampling, and the additional covariance model is very small.
> > > >
> > > > > Of course your goal is not to introduce further errors (I mean, hopefully :D). I was just worried about the effect of bad diagonal estimation network. If the latter has not sufficient capacity to match the optimal diagonal, even from perfect score learning, this could bring some errors (then the covariance could not be 'aligned' with the learned score). But I strongly agree and encourage methods like yours which seek for the true covariance and not heuristics.
> > > > >
> > > >
> > > > **ANSWER:** Thanks for your appreciation of our method. We also strongly agree with your intuition that a bad diagonal estimation network could bring errors.
> > > >
> > > > > What about figure 1? Will you change it or put something more explicit? It's still hard to catch the message without GT comparison. Also I pointed in my Spotted typos a problem with M and K. But if I understand correctly, is the number of timesteps. Then what is the error? The caption of Figure 1 or the K which should be M?
> > > > From what I see, I imagine that you decrease the noise as you go to the right of the figure. Then it's not with different Rademacher samples but different noise levels. Am I missing something here?
> > > > >
> > > >
> > > > **ANSWER:** Apologies for the confusion. You are correct, it should be M. We will correct this in the revision. Essentially, as you can see, increasing the number of Rademacher samples leads to a clearer structure of the diagonal covariance, meaning that we have a better approximation.

---

> > ### Comment · Reviewer_mSjs · 2024-11-21
> > **Thanks (part 2)**
> >
> > 1) I agree the estimator is unbiased, but as you mentioned, the number of samples will affect the variance of the estimate. Then I will add errors to your tables (i.e. report the mean +/- std over several runs) if you have time. I hope the latter will decrease more explicitly with $M$.
> >
> > 2) Sorry, it was not clear to me that you used the true covariance in Figure 3. Since you predict the diagonal from the score using an additional network, I thought you used the true score to learn the diagonal in (a) and the learned one in (b). It's nice that you've added the true covariance (why is it not closer to 0 at the lowest number of steps? You need more samples?).

---

> > > ### Author Response · Authors · 2024-11-23
> > > **Follow-up Response (Part 2)**
> > >
> > > > I agree the estimator is unbiased, but as you mentioned, the number of samples will affect the variance of the estimate. Then I will add errors to your tables (i.e. report the mean +/- std over several runs) if you have time. I hope the latter will decrease more explicitly with M.
> > > >
> > >
> > > **ANSWER:** Thank you for your suggestion. We are considering including the standard deviation in the camera-ready version. From our current observations, the standard deviation of performance in these varying M experiments is very small (less than 0.01). Additionally, Table 9 in the appendix provides the mean and standard deviation results, which further demonstrate that the variance of the FID and NLL metrics is minimal.
> > >
> > > > Sorry, it was not clear to me that you used the true covariance in Figure 3. Since you predict the diagonal from the score using an additional network, I thought you used the true score to learn the diagonal in (a) and the learned one in (b). It's nice that you've added the true covariance (why is it not closer to 0 at the lowest number of steps? You need more samples?).
> > > >
> > >
> > > **ANSWER:** Thanks for the comment. Yes, we use the true score to learn the diagonal in Figure 3(a) and the learned one in Figure 3(b). The poor performance of the true covariance at steps=2 is due to the inconsistency between the mean and covariance, where we used the learned mean and true covariance. We have now updated the figure to use both the true mean and true covariance (please refer to the [DDPM results](https://imgur.com/a/ddpm-mmd-v-s-steps-PlQms0W) and [DDIM results](https://imgur.com/a/ddim-mmd-v-s-steps-fDS0TRZ)). The updated results show that using the true covariance leads to better performance.

---

> > > > ### Comment · Reviewer_mSjs · 2024-11-23
> > > > **Final response**
> > > >
> > > > My following response items overlap as I wrote them following your answers. You do not need to answer me this time as I'm already convinced. You did a great job! Please acknowledge my comments and see them as last suggestions.
> > > >
> > > > - Thanks for the 40 MoG example (preliminary visualizations are thrilling). It's good to show that you acknowledge the limitations of diagonal covariance only while still motivating it by the proximity to the full covariance results.
> > > > - I prefer your log-scale version for Figure 3(c) I find it easier to read (otherwise, half of the x-axis is useless in the non-log version as all methods looks confounded). However, the legend would need to be smaller!
> > > > - Nice for the DDIM-DDPM figure update! (Despite the pointed y log-scale). I find this change way better! It kinda motivates your method analytically as we see that, when we have access to the GT, using the diagonal nearly matches perfectly the use of the full covariance. Of course, it depends on the problem, but it's much appealing as so! Great work. Once again, the log scale might allow to distinguish GT-full from GT-diagonal, especially for DDIM where the difference is indistinguishable. Otherwise, please explicit in the caption (at least) that they are confounded.

---

> ### Author Response · Authors · 2024-11-19
> **Response to Reviewer (Part 4)**
>
> > In equation (10), you still need $H(\tilde{x})$ which can be large as you mention. Do you model it explicitly for training the covariance network or do you model it implicitly using jvp/vjp benefits? This should be discussed. How does it slow down training compared to classical denoising-score matching or other methods?
> >
>
> **ANSWER:** Thanks for raising this. We compute the Hessian $H(\tilde{x})$ implicitly using the Jacobian-vector product. Specifically, for a given sample, we first evaluate its score using the score network $s(x_t)$, then perform a dot product $s(x_t) \cdot v$ to obtain a scalar. Next, we use auto-diff to compute $H(\tilde{x}_t)v$. Finally, we apply the element-wise product to obtain $v\odot H(v_t)v$. This approach allows us to avoid explicitly computing $H(\tilde{x})$  which significantly reduces the training memory and cost. We will make it clear in the revision.
>
> It is important to note that we only train a small additional network, meaning the training cost is comparable to that of classical denoising-score matching. However, since we need to compute the Jacobian-vector product, the computational cost of our method is higher than that of other baselines. Specifically, we summarize the average wall-clock time per iteration for each method below:
>
> |  | DDPM | SN-DDPM | OCM-DDPM |
> | --- | --- | --- | --- |
> | Wall-Clock Time (s/itr) | 0.436 | 0.206 | 0.371 |
>
> > Can the introduction of an additional network (meaning additional approximation) lead to cumulative errors? This is not discussed at all, but the network not being well aligned with the covariance of the trained diffusion will surely lead to poor performances even if the diffusion model was good.
> >
>
> **ANSWER:**
>
> Thank you for this insightful comment. The introduction of the additional diagonal covariance network is indeed a widely adopted approach in popular diffusion frameworks such as I-DDPM and SN-DDPM.  We want to highlight that the introduced network is actually used to  learn to  reduce the covariance estimation error with the following points:
>
> 1. **Comparison to DDPM/DDIM**: In models like DDPM and DDIM, covariance is typically chosen heuristically to be the lower or upper bound of the true variance. In contrast, learning the covariance through an additional network can reduce the approximation error compared to just using the bound.
> 2. **Comparison to Independent Isotropic Variance**: The learned $x_t$-dependent diagonal covariance is significantly more flexible than independent isotropic variance,  allowing lower estimation error.
> 3. **Comparison to Other Diagonal Covariance Learning Methods**: We show that our proposed objective leads to lower estimation errors compared to other methods for diagonal covariance learning. This reduction in error directly translates to improved generative and likelihood results.
>
> Thus, the additional network is introduced not to add approximation error but to actively learn and reduce it, outperforming other heuristic or less flexible methods. Sorry for the misleading on this, we will make this point clear in the revised paper.
>
> > Spotted typos
> >
> **ANSWER:**
> We sincerely thank the reviewer for identifying the typos, which we will address in the revised version. We deeply appreciate the time and thoughtful feedback provided, as it has significantly contributed to improving our work. We kindly request the reviewer to consider our additional explanations and let us know if they address the concerns raised, potentially influencing the assessment of our work. Please do not hesitate to highlight any other points that you believe may be relevant.

---

> > ### Comment · Reviewer_mSjs · 2024-11-21
> > **Thanks (part 4)**
> >
> > 1) What do you mean *per iteration* for your table? It is per training iteration? It's expected and the time complexity is reasonably above, but thanks to explicit that! Would also be nice to see something similar but at inference (where you do not need jvp anymore).
> >
> > 2) Of course your goal is not to introduce further errors (I mean, hopefully :D). I was just worried about the effect of bad diagonal estimation network. If the latter has not sufficient capacity to match the optimal diagonal, even from perfect score learning, this could bring some errors (then the covariance could not be 'aligned' with the learned score). But I strongly agree and encourage methods like yours which seek for the true covariance and not heuristics.

---

> ### Comment · Reviewer_mSjs · 2024-11-21
> **Figure 1**
>
> Sorry, I forgot to ask.
>
> What about figure 1? Will you change it or put something more explicit? It's still hard to catch the message without GT comparison.
> Also I pointed in my **Spotted typos** a problem with $M$ and $K$. But if I understand correctly, $K$ is the number of timesteps. Then what is the error? The caption of Figure 1 or the $K$ which should be $M$?
>
> From what I see, I imagine that you decrease the noise as you go to the right of the figure. Then it's not *with different Rademacher samples* but *different noise levels*. Am I missing something here?

---

> > ### Author Response · Authors · 2024-11-21
> > **Thanks for your detailed response**
> >
> > Hi reviewer mSjs,
> >
> > Thanks for your detailed response. It is very encouraging for us. We are currently working on the revision and will include a new toy experiment that uses the full true Gaussian. Here is a quick visualization of the [training data](https://imgur.com/a/40-mog-RiSq2pU) along with [the results](https://imgur.com/a/mmd-results-using-true-score-covariance-of-40-mog-hNyQ8Lk) of using the true score and true covariance. As you can see, learning a diagonal covariance indeed improves performance compared to using a fixed isotropic covariance. Additionally, using the full covariance further improves the results compared to the diagonal covariance. We find this very promising and believe it will provide valuable insights. We are still working on implementing our method and other baselines, but we want to quickly sync up and share this exciting update.
> >
> > Regarding Figure 1, you are correct—it should be M. We will address your other follow-up questions and upload the revision soon. Thanks again for your detailed comments!

---

> ### Author Response · Authors · 2024-11-23
> **Thanks for your suggestions**
>
> Thanks for your super quick reply. We will include your suggestions in the revision. While we are still working on it, the revised version will be uploaded soon. Thanks again for your suggestion. It has been incredibly helpful in improving our work!

---

### Author Response · Authors · 2024-11-25
**Summary of the Revision**

We sincerely thank all reviewers for their favour of our work and deeply appreciate the time and thoughtful feedback provided, as it has significantly contributed to improving our work. Now the revised version is ready. We summarise the main revision in the following:

- As the reviewer **mSjs** and **tC1T** suggested, we have included a study of the impact of the numbers of Rademacher samples in Appendix C.3.
- As the reviewer **mSjs** and **tC1T** suggested, we have included a toy experiment in Appendix C.1 to highlight the benefit of using full covariance.
- As the reviewer **mSjs**, **tC1T** and **hfvy** suggested, we have included the discussion of potential future work in Section 6.
- As all reviewers suggested, we have included an additional visualisation in Figure 5 to highlight the better trade-off between FID and NLL that the proposed method can achieve.
- As the reviewer **mSjs** and **tC1T** suggested, we have updated the title to “Optimal Diagonal Covariance Matching,” corrected the typos, and revised some narratives as suggested by tC1T to improve the paper's readability.

Once again, we sincerely appreciate the constructive feedback you’ve provided. We would be glad to hear any further suggestions you might have that could help improve our work.

---

> ### Comment · Reviewer_mSjs · 2024-11-26
> **Thanks for the revised version!**
>
> Thanks for your revised version!
>
> I have a last question that came to mind.
>
> Why don't you use the same network to output both $\mathbb{E}[x\mid x_t]$ and $diag(\mathbb{V}[x\mid x_t])$ as you would do for classical parameterized Gaussian model? This could have the benefit of sharing weights, reducing even more the computational overhead of the covariance estimation and potentially limiting mismatch between predicted diagonal and diagonal from the score.
>
> However, I agree that an additional network allow to plug your method to an existing pre-trained denoiser, adding external component (the second network) that will not change the original denoiser architecture. I also agree that this could slow down a bit the training if the shared backbone is large. However, at inference, I only see advantages. Can you comment on that, especially for diffusion model trained from scratch?

---

> ### Author Response · Authors · 2024-11-26
>
> Thanks for the comment. It offers a very insightful suggestion! Our implementation aligns with the architecture you described. As shown in Equation 20, we use a BaseNet as the shared backbone and add an additional head to output $diag(\mathbb{V}[x|x_t])$. The backbone and the mean head $\mathbb{E}[x|x_t]$ are derived from the pretrained diffusion model (see Table 6 in the Appendix for details), with their parameters fixed. We only train the diagonal Hessian head. This architecture also aligns with the design used in the baselines.
>
> As you pointed out, this approach benefits from weight sharing. Additionally, it preserves the original prediction of the mean $\mathbb{E}[x|x_t]$ and the additional inference time is negligible (see Table 4). Furthermore, since the diagonal Hessian head is relatively lightweight, its training cost is even lower than that of training the original diffusion model (see [the wall clock comparison](https://openreview.net/forum?id=fV0t65OBUu&noteId=7tvwnbYHR9) in our previous discussion).

---

### Meta-Review · Area_Chair_foMv · 2024-12-21

**Metareview:**

Two methods are proposed to estimate the diagonal covariance matrices for probabilistic diffusion models, based on optimal covariance matching.

Assuming the covariance matrix to be diagonal is quite restrictive and, as one of the reviewers pointed out, is contradictive to the title "covariance matching".

**Additional Comments On Reviewer Discussion:**

Although the ratings are all quite high, the reviewers have pointed out some major weaknesses, such as:
1. The assumption that the covariance matrix is diagonal may be impractical.
2. Experimental results are not very impressive.
3. Contribution and novelty seem to be slim

---

### Decision · Program_Chairs · 2025-01-22

Accept (Oral)